# Collective colony growth is optimized by branching pattern formation in *Pseudomonas aeruginosa*

Nan Luo[1] ⓘ, Shangying Wang[1] ⓘ, Jia Lu[1], Xiaoyi Ouyang[2] ⓘ & Lingchong You[1,3,4,*] ⓘ

## Abstract

**Branching pattern formation is common in many microbes. Extensive studies have focused on addressing how such patterns emerge from local cell–cell and cell–environment interactions. However, little is known about whether and to what extent these patterns play a physiological role. Here, we consider the colonization of bacteria as an optimization problem to find the colony patterns that maximize colony growth efficiency under different environmental conditions. We demonstrate that *Pseudomonas aeruginosa* colonies develop branching patterns with characteristics comparable to the prediction of modeling; for example, colonies form thin branches in a nutrient-poor environment. Hence, the formation of branching patterns represents an optimal strategy for the growth of *Pseudomonas aeruginosa* colonies. The quantitative relationship between colony patterns and growth conditions enables us to develop a coarse-grained model to predict diverse colony patterns under more complex conditions, which we validated experimentally. Our results offer new insights into branching pattern formation as a problem-solving social behavior in microbes and enable fast and accurate predictions of complex spatial patterns in branching colonies.**

**Keywords** bacterial colony; branching pattern; coarse-grained modeling; optimization model; pattern formation
**Subject Categories** Biotechnology & Synthetic Biology; Microbiology, Virology & Host Pathogen Interaction
**Mol Syst Biol. (2021) 17: e10089**

## Introduction

Self-organized pattern formation is ubiquitous in biology at massively different time and length scales. Examples include the formation of the division ring in a single bacterium (~10 nm) (Weiss, 2004), patterns and structures in a microbial colony (~1 mm–0.01 m) (Ben-Jacob *et al*, 2004), the establishment of body plan during embryonic development (~0.1 mm–0.1 m) (Arthur, 1997), skin patterns of animals (~0.01 m–1 m) (Koch & Meinhardt, 1994), and vegetation patterns in ecological systems (~0.1 m–100 m) (Meron *et al*, 2004). To date, however, our ability to explain, generate, and predictably control self-organized pattern formation has been limited. For instance, in the past 20 years, research in synthetic biology has generated thousands of gene circuits able to program logic operations, switching dynamics, and oscillations (Becskei & Serrano, 2000; Gardner *et al*, 2000; Atkinson *et al*, 2003; Kramer *et al*, 2004; Fung *et al*, 2005). In contrast, only a handful of gene circuits have been engineered to program living cells to generate self-organized patterns (Liu *et al*, 2011; Cao *et al*, 2016; Karig *et al*, 2018; Sekine *et al*, 2018), which are dwarfed in complexity and sophistication by patterns found in nature.

In quantitative analyses of pattern formation (in natural or synthetic systems), the major focus has been placed on a bottom-up approach—i.e., the explanation or prediction of the emergence of patterns from underlying, molecular- and cellular-level interactions. However, the lack of understanding of sufficient mechanistic details can limit the effectiveness of this approach. In contrast, a complementary perspective is to consider the potential physiological implications of such patterns, which are particularly relevant for those formed during microbial colony development. It has been well-recognized that the formation of colonial patterns represents a social behavior in the sense that they emerge from communication and cooperation between individual cells (Ben-Jacob *et al*, 2004). Moreover, it has been speculated that the emergent patterns per se can facilitate the collective survival of the population in a given environment (Kempes *et al*, 2014; Ratzke & Gore, 2016). In other words, the emergence of patterns could reflect a problem-solving capability by a microbial population. If so, the need or the tendency to optimize survival could impose a top-down, physiological constraint on the permissible molecular mechanisms underlying the pattern formation. This approach has been used to analyze the roles of patterns or structures for several biological systems (Honda & Fisher, 1978; Honda & Fisher, 1979; Niklas, 1994; Ho *et al*, 2004). More broadly, the ability of a biological system to optimize survival in a spatial domain may inspire or guide the engineering of patterns or structures beyond biology (Tero *et al*, 2010).

However, the speculated survival role of pattern formation has not been experimentally established. To examine this notion, here

1  Department of Biomedical Engineering, Duke University, Durham, NC, USA
2  School of Physics, Peking University, Beijing, China
3  Center for Genomic and Computational Biology, Duke University, Durham, NC, USA
4  Department of Molecular Genetics and Microbiology, Duke University School of Medicine, Durham, NC, USA
  *Corresponding author. Tel: +1 919 660 8408; Fax: +1 919 668 0795; E-mail: you@duke.edu

we study the formation of branching patterns, which emerge in many microbes when growing on solid surfaces. Examples include bacterial species such as *Pseudomonas* and *Bacillus* (Fig 1A), fungi, slime molds, and lichen (Sumner, 2001; Ben-Jacob *et al*, 2004; Tero *et al*, 2010; Tronnolone *et al*, 2018). Since branching patterns in microbial colonies often emerge under nutrient-deprived conditions (Shimada *et al*, 2004), it is plausible that branched colony growth may represent a survival strategy for microbes. Modeling analyses to date have focused on addressing how such patterns may emerge due to the instability that arises from cell–cell and cell-substrate interactions, including cell growth and movement (Ben-Jacob *et al*, 1994; Kawasaki *et al*, 1997; Matsushita *et al*, 1998b; Kozlovsky *et al*, 1999; Mimura *et al*, 2000; Farrell *et al*, 2013; Giverso *et al*, 2015a; Trinschek *et al*, 2018). However, studies on how branching patterns relate to population fitness have been scarce apart from speculations that branching patterns may optimize resource transport due to increased surface area, as do the branching vessels in mammalian circulatory and respiratory systems (Deng *et al*, 2014).

Here, we approach this problem by considering the colonization of bacteria as an optimization problem: finding the optimal pattern to maximize colony growth efficiency given the growth conditions. The observed patterns formed by *Pseudomonas* colonies in swarming assays (Caiazza *et al*, 2005; Tremblay & Deziel, 2008; Xavier *et al*, 2011) are consistent with the optimal solutions to an optimization model for colony growth under varying conditions. In particular, thin branches are optimal when colonies grow in nutrient-deprived or expansion-limited environments. These results allow us to deduce an optimization rule of colony pattern formation: Colonies develop patterns that maximize the efficiency of colony growth under a particular local environment. Despite a limited understanding of the mechanistic details, this simple rule of pattern formation

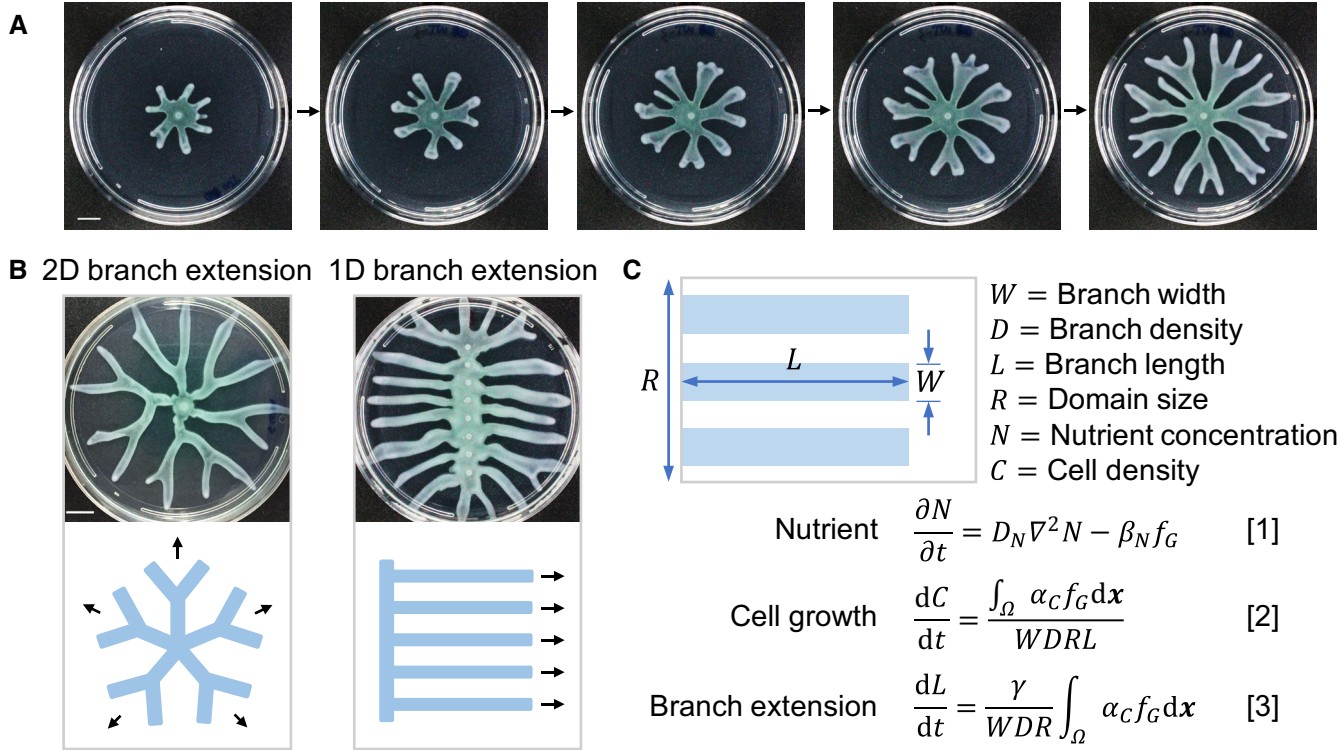

**Figure 1. An optimization model for branched colony growth.**

A   *Pseudomonas aeruginosa* colonies develop branching patterns from a symmetric initial shape when growing on swarming media (recipe described in Xavier *et al*, 2011) in 90-mm petri dishes. Scale bar: 1 cm.

B   *P. aeruginosa* colonies expand in 2D when initiating from a point inoculation (left) but form stripe patterns that extend in 1D when initiating from a strip (right). Top: images of colonies growing on swarming media in 90-mm petri dishes; bottom: schematics of the colony patterns (blue strips represent branches of colonies, and arrows show the directions of branch extension); patterns expanding in 2D can be simplified into bifurcating branches, and patterns expanding in 1D can be simplified to parallel strips. Scale bar: 1 cm.

C   A simple model to describe colony growth with predefined patterns assuming 1D branch extension. The geometry of the colony is given by predefined branch width ($W$) and density ($D$) on the vertical direction and the variable, branch length ($L$), on the horizontal direction (shown in the schematic; blue strips represent branches of a colony growing from one end of a rectangular domain). $R$ is the domain size on the vertical direction. The diffusion and consumption of the nutrient ($N$) is described by Eq [1], where $D_N$ is the diffusivity and $\beta_N$ is the consumption rate of nutrient. Eq [2] describes the growth of cells ($C$), and $\alpha_C$ is the cell growth rate. The cell growth function ($f_G$) is a function of $N$ and $C$. The total amount of cell growth in the colony, $\Omega$, is averaged to the total colony area. The elongation rate of branches is obtained based on the assumption that the net expansion of the colony is proportional to the amount of cell growth (Eq [3]), and $\gamma$ is the efficiency of colony expansion (see Appendix Supplementary Methods for further details).

Source data are available online for this figure.

provides a new way of predicting complex patterns of *Pseudomonas* colonies under various conditions, laying the foundation for rational programming or de novo generation of bacterial colony patterns for engineering applications (Liu *et al,* 2011; Cao *et al,* 2016; Cao *et al,* 2017; Karig *et al,* 2018; Sekine *et al,* 2018).

# Results

### An optimization model of branched colony growth

In contrast to typical biophysical models, we treat the formation of colony patterns as an optimization problem using a coarse-grained model. In particular, we ask how the overall colony growth depends on critical parameters that define branching patterns, *regardless of how these patterns emerge from molecular-level interactions*. In our model, the pattern of a colony is constrained by given parameters and confines cell growth and cell movement. We vary the colony pattern to investigate how patterns affect the objective function: the efficiency of biomass accumulation of the colony.

In swarming assays, *Pseudomonas* colonies typically initiate from point inoculations and form branches that extend in two dimensions (2D) (Fig 1A). As the colony expands, bifurcations of branches result in an approximately constant partitioning of the swarming medium for each branch. For the convenience of modeling, we first consider a simpler scenario of branch extension: When initiated from a strip, colonies form parallel, non-bifurcating branches extending from one end of a rectangular plate to the other (Fig 1B). This one-dimensional (1D) extension preserves the feature of a constant partitioning of the growth medium.

In this simplified framework, branches extend from one end of the plate to the other. The branching pattern is uniquely defined by two parameters (Fig 1C): the branch width ($W$) and the branch density ($D$, the number of branches in one unit of length). The total width of all branches is $WDR$, and it cannot be greater than the domain size; hence, $WDR \leq R$, or $WD \leq 1$. In the extreme case, when $WD = 1$, the colony becomes non-branching or circular.

We further consider three processes involved in colony growth: the consumption and diffusion of the nutrient, cell growth, and branch extension (Fig 1C). Here, the colony is growing in a rectangular domain and consumes a diffusive nutrient ($N$, Equation 1). As illustrated in Fig 1C, the boundary of the colony is defined by $W$, $D$, which are given constants, and the branch length, $L$, which is a time-dependent variable. The growth of the cells ($C$) is confined within the colony boundary: As cells multiply, biomass accumulates and is uniformly allocated within the entire colony (Equation 2). The cell growth function ($f_G$) is a function of $N$ and $C$ fitted to the growth curves of *Pseudomonas* in liquid swarming media (Appendix Fig S1A). Cell movement is simplified as branch extension (Equation 3): We assume that the total energy extracted from nutrient consumption is conserved and allocated between growth and colony expansion; therefore, the rate of branch extension is proportional to the amount of cell growth (see Appendix Supplementary Methods for details). Here, $\gamma$ is a coefficient relating the amount of energy for cell movement to the expansion rate of the colony. Using the same amount of energy, with greater $\gamma$, the colony expands faster, so we hitherto refer to $\gamma$ as the colony expansion efficiency. We can experimentally change the expansion efficiency

by altering the agar density of the swarming media, as *Pseudomonas* colonies expand faster on media with lower agar densities (Appendix Fig S2).

### Branching patterns enable optimal growth when resource is limited

Using our simple model, we examine how the total colony biomass depends on different combinations of branch width ($W$) and branch density ($D$) for different environmental conditions dictated by other parameters in the model (Fig 2A). For each condition, an optimal combination of $W$ and $D$ exists to maximize biomass accumulation. In particular, when the initial concentration of nutrient ($N_0$) or the expansion efficiency of cells ($\gamma$) is low (Fig 2A, panel a), optimal growth requires a small branch width and a relatively low branch density. The advantage of thin branches over wide ones decreases if colonies grow with abundant nutrients or on surfaces that allow faster expansion. With sufficiently high nutrient concentrations or expansion efficiencies, non-branching colonies accumulate biomass most efficiently (Fig 2A, panels b, c).

To understand why the optimal patterns vary with the growth conditions, we examine the dynamics of the system using the model (Fig EV1). The distribution of nutrient consumption reveals that the utilization of nutrient is mainly at the colony front and edges when nutrient concentration or the expansion efficiency is low, since nutrient is quickly depleted in the colony-covered regions (Fig EV1A). In this case, the total amount of nutrient utilization is correlated with the length of the colony boundaries, which is greater in colonies with thin branches than in non-branching colonies. However, with abundant nutrient or high expansion efficiency, colonies expand before consuming all the nutrient in the area covered by cells (Fig EV1B and C). Hence, the consumption of nutrient is also related to the colony area, which is higher in colonies expanding uniformly.

To experimentally test these predictions, we compared the growth of wild-type *Pseudomonas aeruginosa* PA14 strain and its hyperswarming mutants in swarming assays. We obtained the mutants using experimental evolution as described in van Ditmarsch *et al,* 2013. *Pseudomonas aeruginosa* PA14 was grown on swarming media for 20 h, and the entire colony was collected from the plate. A fraction of the collected cells was inoculated on a new plate with swarming media. We repeated this procedure for seven consecutive days and isolated mutants that do not develop branching patterns but formed irregular or circular colonies, a phenotype called hyperswarming (van Ditmarsch *et al,* 2013). The same as two of the strains reported by van Ditmarsch *et al,* 2013, we identified a single point mutation (V178G) in a flagellar synthesis regulator gene, FleN (PA14_45640) in our hyperswarmer mutants.

Consistent with the model predictions, when grow on swarming media with limited nutrient and high agar density, where the colony expansion efficiency is low (Appendix Fig S2), the wild-type colonies grow significantly better than the non-branching hyperswarmers (Fig 2B, panel a). The disadvantage in colony growth (with an average of 45.8% less biomass accumulation than the wild type) of hyperswarmers cannot be accounted for by the reduction in growth rate, which is 7% lower than that of the wild type and leads to an average of 10.7% reduction in biomass accumulation in liquid media with the same concentration of nutrient (Appendix Fig S1B).

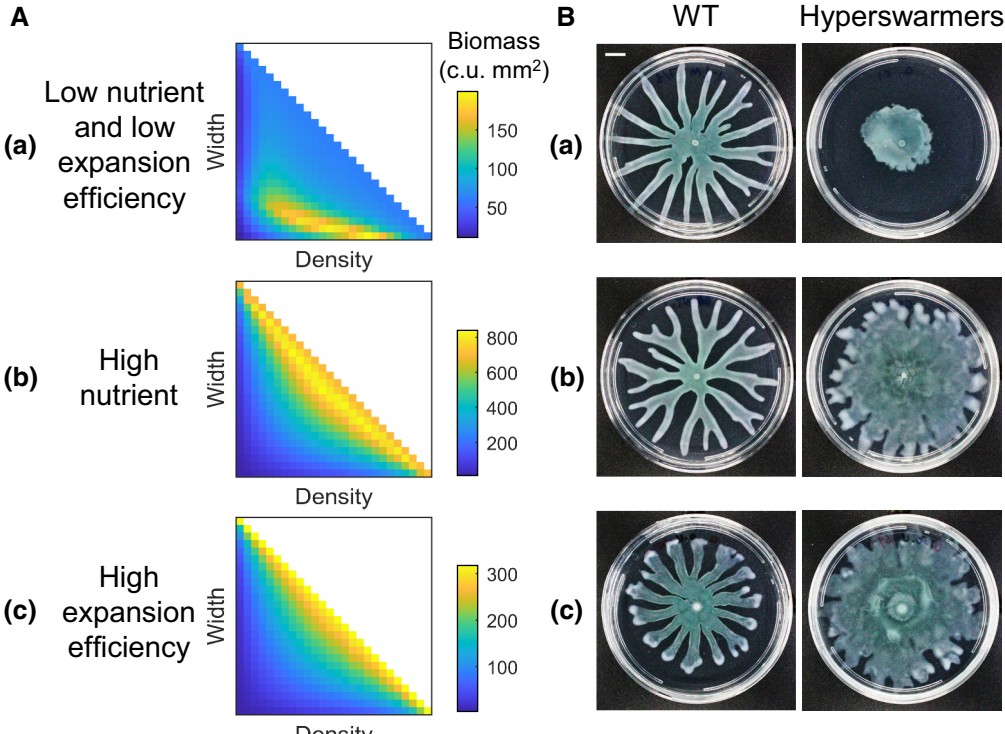

**Figure 2.  Branching patterns enable optimal colony growth when nutrient or expansion of the colony is restricted.**

A   The optimization model implemented with different combinations of branch width and density reveals the optimal colony patterns that yield the highest biomass under different conditions. Colors indicate the total biomass (unit: c.u. mm$^2$; c.u.: cell density unit) at the same time point ($t$ = 24 h). At or beyond the diagonal of the heatmap (where $WD$ = 1), the colony becomes uniform with no branches. (a), (b), and (c) correspond to different initial nutrient concentrations ($N_0$) or expansion efficiency ($\gamma$) (a: $N_0$ = 8 g/l, $\gamma$ = 7.5 mm/h/c.u.; b: $N_0$ = 30 g/l, $\gamma$ = 7.5 mm/h/c.u.; c: $N_0$ = 8 g/l, $\gamma$ = 25 mm/h/c.u.). Other parameters: $D_N$ = 6 mm$^2$/h; $\beta_N$ = 160 g/l/h/c.u.; $\alpha_C$ = 0.8/h; $K_N$ = 0.8 g/l; $C_m$ = 0.05 c.u.

B   *Pseudomonas* colonies with the optimal patterns predicted by the model show higher growth efficiency than the ones with non-optimal patterns. When nutrient or expansion of the colony is restricted (a: growing on media with 4 g/l casmino acids and 0.5% agar; the experiment was independently replicated twice; $n$ = 5 for wild type and $n$ = 10 for hyperswarmers), wild-type *Pseudomonas* PA14 that develop branching patterns grow more efficiently than hyperswarmers that show uniform expansion. On media with higher nutrient concentration (b: with 8 g/l casmino acids and 0.5% agar; the experiment was independently replicated four times; $n$ = 12 for wild-type and $n$ = 24 for hyperswarmers) or wetter surface (c: with 4 g/l casmino acids and 0.4% agar; the experiment was independently replicated twice; $n$ = 4 for wild type and $n$ = 8 for hyperswarmers), hyperswarmers yield more biomass than the wild type. Images were taken at 20 h after inoculation and are representatives of replicates. One image from Fig 1A is reused in (b). Scale bar: 1 cm.

Source data are available online for this figure.

On nutrient-rich media or media with lower agar density, non-branching colonies expand and grow more efficiently than the branching ones, as predicted by the model (Fig 2B, panels b, c).

Our modeling analysis further shows that the optimal combination of $W$ and $D$ can be tuned by changing the growth environment, e.g., by varying the nutrient availability ($N_0$) and colony expansion efficiency ($\gamma$) (Fig 3A). The observed patterns of wild-type *Pseudomonas* colonies vary with the compositions of the swarming media, and the trend roughly follows the predictions of the model: as the nutrient concentration increases or the agar density decreases, *Pseudomonas* form colonies with wider branches and greater total colony areas (Figs 3B and EV2).

### Predicting branching patterns using the optimization rule

Our results suggest that the branched colony growth in PA14 on swarming media approximately follows an optimization rule (for the range of experimental conditions examined): Under each condition,

bacterial colonies develop the optimal pattern that maximizes growth efficiency. This simple rule imposes a constraint on the branching process, which enables the prediction of colony patterns as experimental conditions change. Again, the initiation of branches in this model is imposed by defining the branch widths and densities. In particular, results from screening (Fig 3A) provide a mapping between the growth conditions and the optimal combinations of $W$ and $D$. If we assume that this mapping is maintained locally, it allows us to determine the local $W$ and $D$ of colony branches and predict the colony patterns under more complex conditions.

To test this notion, we first relaxed the simplifying assumption of 1D branch extension. Instead, we implemented a model of branching growth in 2D starting from one (Fig 4A) or multiple inoculating points. Each branch extends following the local nutrient gradient; the boundary of a branch is set by $W$, and bifurcations are determined by $D$ (Figs 4A and EV3). Specifically, we track the tip of a growing branch and calculate the local branch density. As the branch extends, if the local branch density falls below a predefined threshold

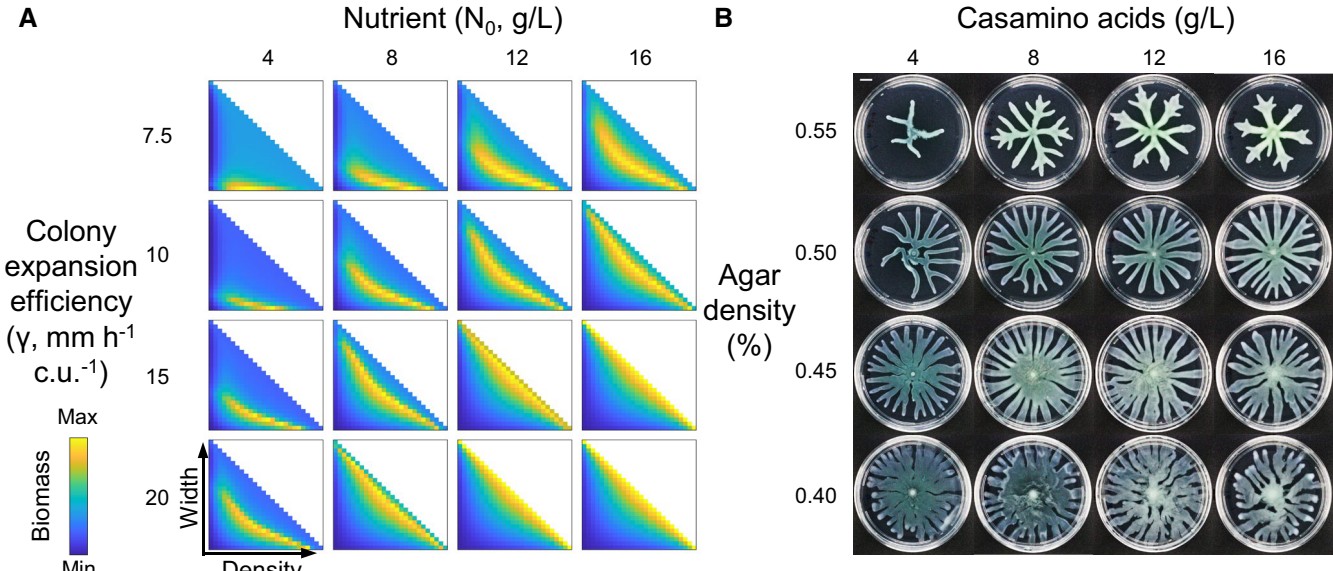

**Figure 3. The optimal colony patterns that maximize biomass accumulation vary with growth conditions.**

A Under various sets of initial nutrient concentrations and expansion efficiencies, the optimal branch density and width that yield the highest biomass varies. Colors indicate the total biomass (unit: c.u. mm$^2$) at the same time point ($t$ = 24 h) scaled to the min/max values in each subpanel. At or beyond the diagonal of the heatmap (where $WD$ = 1), the colony becomes uniform with no branches. Other parameters: $D_N$ = 6 mm$^2$/h; $\beta_N$ = 160 g/l/h/c.u.; $\alpha_C$ = 0.8/h; $K_N$ = 0.8 g/l; $C_m$ = 0.05 c.u.

B The features of the patterns developed by *Pseudomonas* colonies under different growth conditions are generally consistent with the optimal patterns predicted by the model. The colony expansion efficiency is modulated by changing the agar density. Images of colonies were taken once colonies reached the plate boundaries or stopped growing. The experiment was independently replicated five times. Images shown are representatives of the replicates. Scale bar: 1 cm.

Source data are available online for this figure.

($2/3D$), branch bifurcation is triggered and a new sub-branch initiates. The predictions of the optimal patterns by models with 1D or 2D branch extension formulations are consistent (Appendix Fig S3).

A key new assumption is that $W$ and $D$ are set according to the local growth environment, which depends on the initial distributions of nutrient and changes as the colony expands. Starting from a single inoculating point on an initially uniform distribution of growth medium, the resulting patterns under different conditions reliably capture the branching patterns observed experimentally in terms of both branch structure and distribution (Fig 4B).

The quantitative features of these patterns also depend on other model parameters, such as the diffusivity of the nutrient ($D_N$) and the growth rate of cells ($\alpha_C$). Varying these parameters leads to diverse branching patterns, including thin dendrites or patterns with densely packed fingers (Fig EV4). These patterns resemble colonies of other bacterial species, such as *Pseudomonas dendritiformis* (Ben-Jacob *et al*, 2004). Therefore, our model is applicable to other microbial systems and may explain why different bacterial colonies generate drastically distinctive patterns.

This modeling framework also enables interpretation or prediction of pattern formation under heterogeneous conditions. As an illustration, consider the branched colony growth on a swarming medium with a linear gradient of nutrient. When the gradient is steep enough, we observe asymmetric colony growth, but, unexpectedly, colonies expand faster on the side with lower nutrient concentration (Fig 5A). With appropriate parameters, our model is able to reproduce this counterintuitive phenomenon (Fig 5A). Simulations suggest that

the phenomenon results from the asymmetry of $W$ and $D$: Limitation of nutrient triggers the development of thin branches as opposed to the wider branches on the nutrient-rich side; with comparable biomass accumulation rates on each side, the thinner branches extend faster, leading to apparent anti-gradient expansion.

To explore the parameter space that allows anti-nutrient-gradient growth, we carried out systematic parameter screening (Fig 5B). This scheme can be used to search parameters for simulating more complex patterns, such as patterns of colonies growing on media with arbitrary patterns of nutrients or surface wetness. Given one set of model parameters, firstly we find out the optimal patterns under different nutrient concentrations by screening through combinations of $W$ and $D$ (Step 1), thereby obtaining the mapping between the nutrient concentrations and the optimal patterns (Step 2). By implementing this mapping in the model, we can predict the colony patterns growing under heterogeneous distributions of nutrient and search for parameters that generate anti-gradient growth (Step 3).

To accelerate the screening process and explore the parameter space more extensively, we used machine learning to emulate the model's predictions (Step 4). Step 1 is very time-consuming because for each parameter set, searching for the optimal pattern requires screening through thousands of patterns. A neural network-based framework we developed previously (Wang *et al*, 2019) allows us to accelerate the process more than 30,000 times with high accuracy (Appendix Fig S4).

Using the neural networks trained with the results of 1,000 parameter sets, we screened 30,000 parameter sets in the parameter

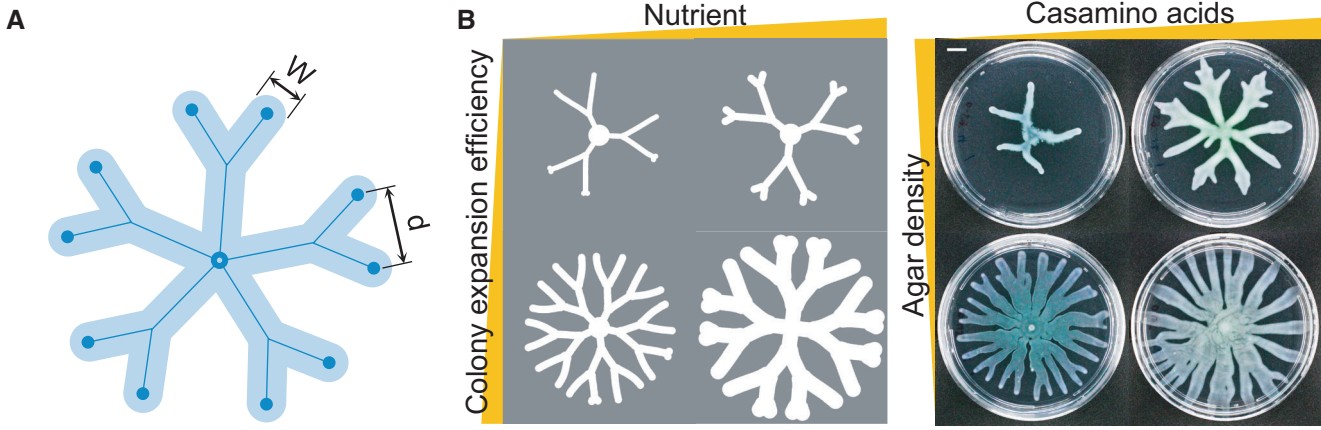

**Figure 4. Predicting 2D colony patterns by applying the optimization rule.**

A  Simulating 2D expansion of branching patterns. The colony starts from a point inoculum (open circle) where branches initiate. Solid lines represent the trajectories of the branch tips (dots). The local branch density at a certain branch tip is given by $1/d$, where $d$ is the distance of the branch tip to its nearest neighbor. The given branch width ($W$) and density ($D$) determine the shape and bifurcation of branches: The boundary of the colony is at a radius of $W/2$ from the branch tip trajectories; the branch bifurcates to maintain the local branch density around $D$. The growth direction of branches follows the local nutrient gradient. Nutrient distribution, cell growth, and branch extension are calculated as described earlier (Fig 1C).

B  The simulated colony patterns capture the general features of the observed patterns of *Pseudomonas* colonies. For the simulations, under each condition, we implement the optimal $W$ and $D$ that give the maximum biomass to generate the predicted optimal patterns. Other parameters: $D_N$ = 6.951 mm²/h; $\beta_N$ = 216.9 g/l/h/c.u.; $\alpha_C$ = 1.486/h; $K_N$ = 1.156 g/l; $C_m$ = 0.0548 c.u. In experiments, the colony expansion efficiency is modulated by changing the agar density. Images from Fig 3B are reused. Scale bar: 1 cm.

Source data are available online for this figure.

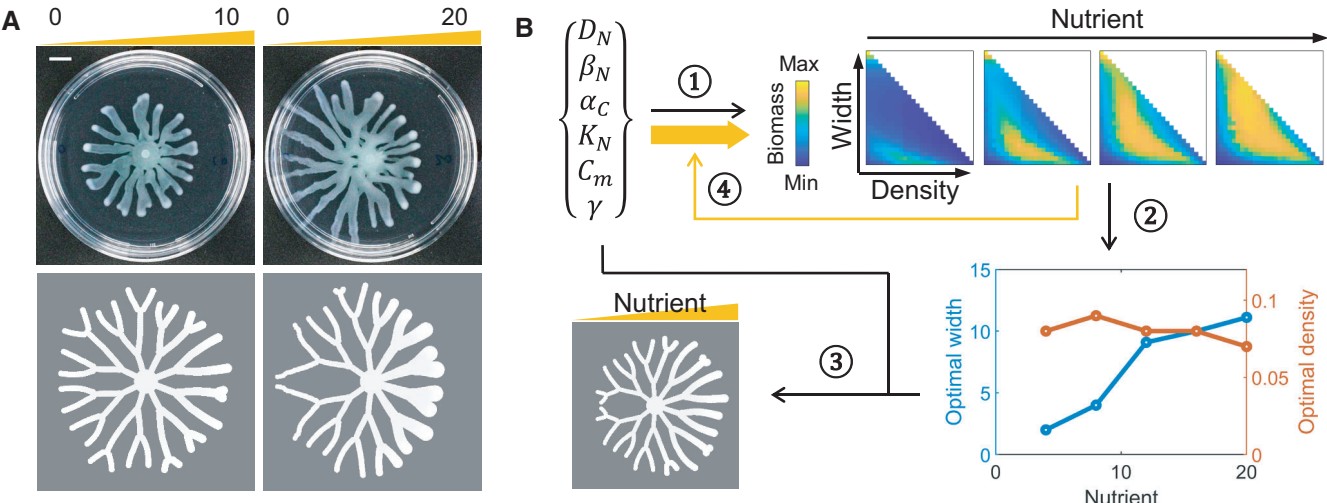

**Figure 5. Predicting colony patterns under heterogeneous conditions.**

A  Anti-nutrient-gradient growth observed in *Pseudomonas* colonies (the upper panels) and simulated using the model (the lower panels). Numbers: casamino acid concentration (g/l) on the two ends of the petri dish. The experiment was independently replicated three times, and representative images are shown. Scale bar: 1 cm. Parameters used in the simulations are obtained by systematic screening described in (B): $D_N$ = 5.749 mm²/h; $\beta_N$ = 195.5 g/l/h/c.u.; $\alpha_C$ = 1.105/h; $K_N$ = 0.6635 g/l; $C_m$ = 0.07890 c.u.; $\gamma$ = 7.513 mm/h/c.u.

B  Systematic screening for the parameter space that satisfies the growth behaviors of colonies under heterogeneous conditions. ① With a given set of randomized model parameters, we generate heatmaps of biomass under different nutrient concentrations by screening through combinations of branch widths and densities (colors represent biomass accumulation; unit: c.u. mm²); ② we find the optimal patterns under different nutrient concentrations and obtain the mapping between the optimal patterns and the nutrient concentration; ③ with the mapping and the model parameters, we predict the patterns under heterogeneous conditions (e.g., media with a nutrient gradient); ④ to accelerate and enhance the throughput of the screening, we use the simulation data of step ① to train neural networks that allow us to emulate the model and generate more data with great efficiency (yellow arrows).

Source data are available online for this figure.

space estimated from experimental measurements (Appendix Supplementary Methods). We identified a parameter space that generates anti-nutrient-gradient colony expansion consistent with experimental observations. The distribution of the identified parameters reveals that for anti-nutrient-gradient growth to occur, the nutrient consumption rate ($\beta_N$) needs to be relatively high and the cell growth rate ($\alpha_C$) and expansion efficiency ($\gamma$) need to be relatively low (Appendix Fig S5), suggesting the association of this phenomenon with constrained cell growth and movement.

The model is also able to predict colony patterns with increasingly complex growth configurations. We seeded multiple colonies in the same petri dish with different configurations, including scattered dots, continuous lines, and complex patterns. Simulations of multiple colonies with these various initial seeding configurations correlate well with observed patterns (Figs 6 and EV5). As branches

of neighboring colonies avoid each other, patterns vary with the configuration of the initial seeding spots. In our model, the lengths of branches are determined by the local biomass accumulation, and the growth directions are guided by the local nutrient gradient. The model captured the characteristics of the growth direction and length of branches starting from different seeding configurations, demonstrating that the local growth is sufficient to account for these growth behaviors.

## Discussion

In typical biophysical models, patterns *emerge from assumed local interactions* with appropriate parameter values. Such an approach is limited by the difficulties in mapping the experimental system to the

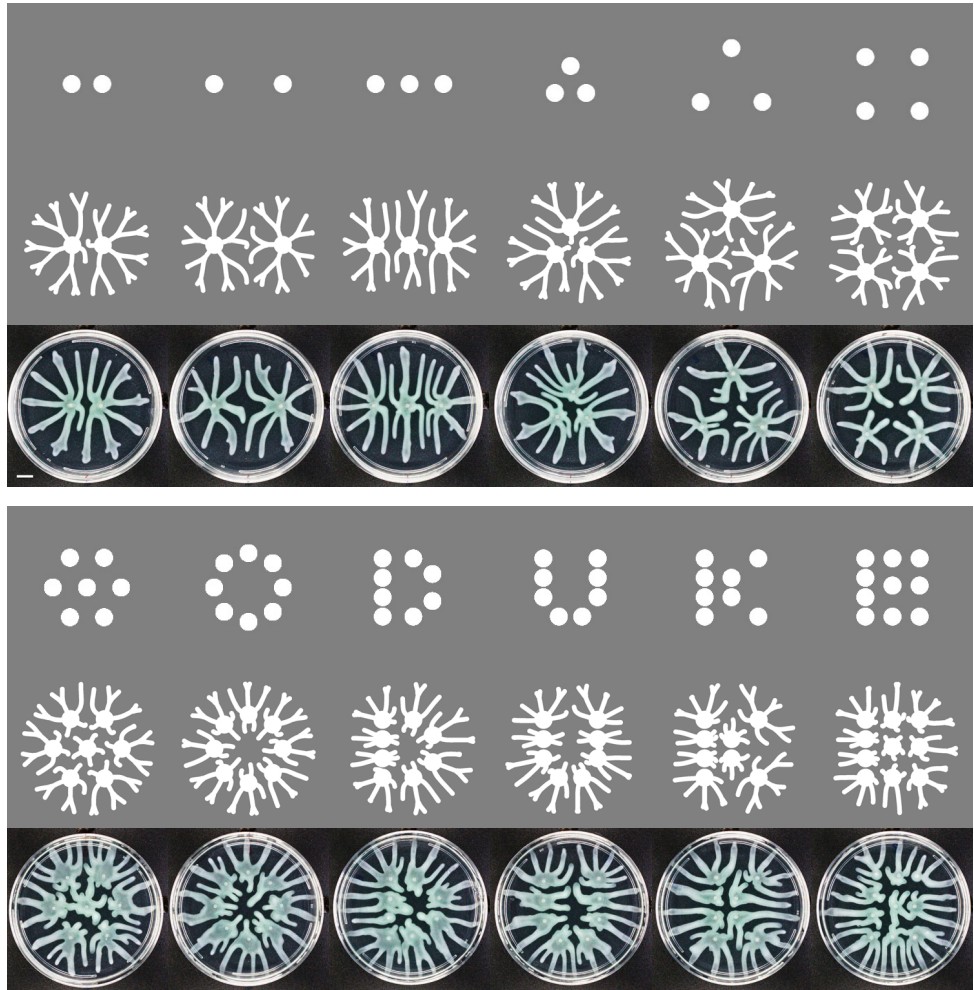

**Figure 6. Predicting colony patterns with various seeding configurations.**

Simulated and observed colony patterns when colonies initiate from discrete dots, continuous lines, or complex patterns that carry information. Upper rows: initial seeding locations; middle rows: simulations based on the optimization rule; lower rows: images of *Pseudomonas* colonies inoculated with the corresponding configurations using an automated liquid handling system (0.1 μl cell culture with $OD_{600}$ ~0.2 was dispensed at each spot). The experiment was independently replicated three times, and representative images are shown. Scale bar: 1 cm. In simulations, patterns are initialized from spots with radius of 5 mm and uniform initial cell density $C_0 = 1.6$ c.u. The parameters for the simulations are as follows: $D_N = 5.749$ mm²/h; $\beta_N = 195.5$ g/l/h/c.u.; $\alpha_C = 1.105$/h; $K_N = 0.6635$ g/l; $Cm = 0.07890$ c.u.; $\gamma = 4$ mm/h/c.u.; $N_0 = 14.5$ g/l.

Source data are available online for this figure.

mathematical description at the proper abstraction level, such that the model can both capture the overall patterning dynamics and guide experimental interrogation. A typical modeling framework is to use reaction-diffusion equations, where "reaction" describes processes such as cell reproduction and death and "diffusion" is dictated by cell dispersal (Kawasaki *et al,* 1997; Matsushita *et al,* 1998a; Kozlovsky *et al,* 1999; Mimura *et al,* 2000; Trinschek *et al,* 2018). In the simplest case of such a model, cell motility is described by a diffusion term, where the dispersal rate constant provides a lumped description of the collective motion of cells (active or being pushed by neighboring cells) (Kawasaki *et al,* 1997; Matsushita *et al,* 1998a; Mimura *et al,* 2000). Models taking these forms have demonstrated the mathematical conditions leading to pattern emergence but have limited predictive power of observed branching morphologies under different conditions (Giverso *et al,* 2015b). More sophisticated descriptions of cell motility have been used, but the invoked mechanistic assumptions are difficult to test experimentally (Kozlovsky *et al,* 1999; Trinschek *et al,* 2018).

To bypass the difficulties in modeling colony branching patterns based on biophysical mechanisms, we approach this problem from a different perspective: whether branching patterns serve a physiological role in bacterial colonies. We frame the growth of colonies as an optimization problem to address the relationship between colony patterns and the fitness of the population. In our modeling framework, the essential properties of branched patterns are imposed, which in turn determine the outcome, the overall biomass accumulation. This approach allows a direct mapping between population fitness and colony morphology. In principle, this modeling framework can also be applied to other systems with similar dynamics of spatial expansion and resource utilization, for example, ecosystems that generate spatial patterns such as vegetation and mussel beds (Rietkerk & van de Koppel, 2008).

The optimization model shows that colonies with thin branches have a higher growth advantage when nutrient access or cell motility is restricted; otherwise, wide branches or non-branching expansion are more efficient for colony growth. These results suggest that the emergence of branching patterns may be an adaptation of bacterial colonies to surface growth in adverse environments where cell growth or motility is constrained. Similar to what we find in bacterial colonies, vegetation patterns ranging from stripes to spots emerge under resource-limited conditions (von Hardenberg *et al,* 2001; Rietkerk & van de Koppel, 2008), hinting at similar underlying mechanisms. Beyond the colony geometry, other physiological parameters, such as the cell motility, the cell growth rate, and the balance between motility and growth, may also be subject to

evolutionary optimization (van Ditmarsch *et al,* 2013; Fraebel *et al,* 2017; Gude *et al,* 2020). Our findings suggest that evolution may select for the optimal colony morphology, which is a macroscopic, population-level property resulting from cellular-level interactions. The formation of multicellular and macroscopic structures by microbial communities that benefit the population as a whole is widely observed; other examples include the formation of patches (Ratzke & Gore, 2016), filaments (Pfeffer *et al,* 2012), spore-filled fruiting bodies (Munoz-Dorado *et al,* 2016), and biofilms with intricate structures (Epstein *et al,* 2011; Wilking *et al,* 2013; Kempes *et al,* 2014; Gingichashvili *et al,* 2019). These social traits of microbes reflect intercellular cooperation and coordination, which are the hallmarks of multicellularity (Shapiro, 1988; Lyons & Kolter, 2015). More broadly, the collective problem-solving ability without centralized control by microbes can inspire algorithm design and novel computing methodology (Abelson *et al,* 2000; Tero *et al,* 2010).

Due to the complex and tangled nature of biological systems, mathematical descriptions that have both high accuracy and generality are difficult to obtain. By choosing an appropriate abstraction level, however, one can often obtain simple and general rules that have quantitative predictive power. Examples include the prediction of collective behavior of a population using coarse-grained abstraction of local interactions (Maire & Youk, 2015; Gordon, 2016), unifying rule that predicts outcomes of mutualistic systems (Wu *et al,* 2019), the linear correlation between cell growth and lysis (Lee *et al,* 2018), the interdependence of gene expression and cell growth or size (Scott *et al,* 2010; Tanouchi *et al,* 2015), scaling laws of drug responses (Wood *et al,* 2014), and prediction of complex community dynamics based on pairwise interactions (Friedman *et al,* 2017; Venturelli *et al,* 2018). This coarse-graining strategy has also proven effective in guiding the measurement and tuning of synthetic gene circuits (Shin *et al,* 2020).

By abstracting away low-level mechanistic details of branch formation, our coarse-grained description of the branching dynamics allows us to deduce a simple rule of colony pattern formation that is experimentally validated: Colonies form patterns that maximize the growth efficiency in the particular growth environment. This simple rule imposes a constraint for the molecular interactions underlying branching pattern formation: That is, the underlying biophysical models should have the ability to generate the branching widths and densities required for optimal growth. Such a constraint has not been considered in previous models of branching patterns. As demonstrated by our results, this simple rule also enables predictable programming of complex patterns by controlling growth conditions and seeding configurations.

# Materials and Methods

### Reagents and Tools table

| Reagent/Resource | Reference or source | Identifier or catalog number |
|---|---|---|
| **Experimental models** | | |
| *Pseudomonas aeruginosa* PA14 | Joao B. Xavier's lab (Memorial Sloan-Kettering Cancer Center, New York) | N/A |
| **Chemicals, enzymes and other reagents** | | |
| $Na_2HPO_4$ (anhydrous) | Sigma | S3264 |

**Reagents and Tools table**   (continued)

| Reagent/Resource | Reference or source | Identifier or catalog number |
| --- | --- | --- |
| $KH_2PO_4$ (anhydrous) | Sigma | P5655 |
| NaCl | Sigma | S3014 |
| $MgSO_4$ | Sigma | M7506 |
| $CaCl_2$ | Sigma | C1016 |
| Casamino acids | BD | Bacto 223120 |
| Granulated agar | BD | Difco 214530 |
| **Software** | | |
| MATLAB R2019b | MathWorks | |
| **Other** | | |
| MANTIS automated liquid handler | FORMULATRIX | |
| UVP Colony Doc-It Imaging Station | Analytik Jena | |
| Plate reader | Tecan | Infinite 200 |

## Methods and Protocols

### Bacterial strains and growth conditions

The strains used in this study were *Pseudomonas aeruginosa* PA14 (wild-type) and *Pseudomonas aeruginosa* PA14 *fleN* (the hyper-swarmers) isolated from experimental evolution. To grow bacteria colonies and observe the patterns, bacteria were cultured in LB medium overnight in a shaker incubator at 37°C and 200 rpm. The overnight culture (200 μl) was diluted in 1 ml fresh LB medium and incubated at 37°C and 200 rpm for an additional 3 h to allow the cells to recover to the exponential growth stage and reach a final concentration of $OD_{600}$ 0.2–0.4. The swarming medium was freshly prepared (Xavier *et al*, 2011), and 20 ml of medium was pipetted into each petri dish (100 mm, Falcon). After the medium had solidified, 1 μl of cell culture was pipetted onto the medium surface at the center of each plate, and plates were left to dry on the bench for 15 min with lids open. The plates were then incubated upside down at 37°C in an incubator.

We used a MANTIS automated liquid handler (FORMULATRIX) to inoculate multiple colonies with designed initial patterns. The patterns were designed using the MANTIS dispense designer with the template for a 1,536-well microplate. Cell culture (0.1 μl) was dispensed onto the medium surface at each spot of the initial patterns.

### Swarming medium

We prepared swarming medium to generate branching patterns of *Pseudomonas* following the following recipe adapted from Xavier *et al*, (2011): 200 ml of 5× stock phosphate buffer, 1 ml of 1 M $MgSO_4$, 1 ml of 0.1 M $CaCl_2$, casamino acid stock solution (200 g/l), agar stock solution (1.25%, melted), and sterilized water to make up 1 liter. The volumes of casamino acids and agar were determined by the needed final concentrations. To make 1L 5× phosphate buffer stock solution, we dissolved 12 g $Na_2HPO_4$ (anhydrous), 15 g $KH_2PO_4$ (anhydrous), and 2.5 g NaCl in water and sterilized by autoclaving. To make 200 ml casamino acid stock solution, we dissolved 40 g casamino acids (Gibco™ Bacto™ 223120) in water and sterilized by filtering. To make 1L agar stock solution, we added 12.5 g granulated agar (BD Difco™ 214530) in water and sterilized by autoclaving. Each swarming plate was prepared by pipetting

exactly 20 ml of medium into a petri dish (100 mm, Falcon), and the dish was allowed to cool for 20 min to 1 h.

To make swarming plates with a nutrient gradient (for example, 0–20 g/l casamino acids), first we pipetted 10 ml of melted medium containing 20 g/l casamino acids into a petri dish and elevated one side of the dish. An agar wedge formed when the medium solidified. Then, we laid the dish flat, pipetted another 10 ml of melted medium containing no casamino acids, and let it solidify. A concentration gradient of casamino acids formed after a few hours due to the diffusion of casamino acids between the two layers of agar.

### Imaging and quantification of colonies

Bacterial colonies growing on plates were imaged with a UVP Colony Doc-It Imaging Station with epi white light. The brightness and contrast of all images were enhanced using the same setting.

Branch widths of colonies were measured in a semi-automated manner using a custom-made MATLAB code. An algorithm was developed to measure the widths of all branches in a colony at the same radial distance from the colony center; for the same colony, the measurements were repeated at different radial distances which were equally spaced. When the algorithm made mistakes in distinguishing individual branches or introduced errors when measuring branches that were not extending in the radial directions, these data were excluded by manual inspection. The measured branch widths were averaged and recorded as the mean branch width of this colony. The mean branch widths of several colonies under the same condition were averaged to obtain the average branch width of colonies in a particular experimental group.

### Experimental evolution and sequencing

We carried out experimental evolution of *Pseudomonas aeruginosa* PA14 on swarming media with 8 g/l casamino acids and 0.5% agar. After 20 h of growth, we collected the entire colony from the plate and diluted the cells with LB medium. We used approximately $10^6$ cells to inoculate a new plate with swarming medium. This procedure was repeated for seven consecutive days. The collected cell population of day 7 was streaked on a LB plate to isolate individual

strains that emerged during evolution. We observed colony patterns of the isolated strains and carried out whole-genome sequencing.

Genomes of *Pseudomonas aeruginosa* PA14 variants and the ancestor strain (the lab wild-type strain) were sequenced using Illumina NovaSeq (6000 S-prime 150 bp PE) with an average of 300–400 coverage. Initial read quality checks were carried out using FastQC. Sequencing reads were then processed using the TrimGalore toolkit to trim low-quality bases from the 3′ end of the reads and Illumina sequencing adapters. Only reads that were 20nt or longer after trimming were kept for further analysis. Reads were aligned to the Ensembl *Pseudomonas aeruginosa* PA14 genome reference (Pseu_aeru_PA14_V1) using BWA-mem. Putative variants were detected using variant caller FreeBayes and annotated. Only variants for which at least one sample had been genotyped as different from the reference were kept.

### Growth measurement

We measured the growth curves of *Pseudomonas* using a plate reader (Tecan Infinite 200). In each well of a 96-well plate, we added 200 µl liquid swarming media (prepared by substituting the agar solution with water), 2 µl cell culture (overnight cultures of different replicates and strains were diluted 10×–50× to the same cell density), and 50 µl mineral oil (to prevent evaporation). The cells were then incubated in the plate reader at 37°C, and $OD_{600}$ measurements were taken at 10-min intervals for 24 h.

### Measuring the relative mobility of cells

*Pseudomonas* colonies were grown on solid LB media, so they expanded radially without developing branches. After 16 h of growth in 37°C, the diameters of the colonies were measured under a microscope every 2–3 h. The growth speed (the increasing rate of the colony diameter), *v*, of each colony was obtained by fitting the diameters to linear functions of time. Assuming the growth and expansion of a radially expanding colony can be described by Fisher's equation, the cell diffusivity or the relative mobility of cells is proportional to $v^2$ (Murray, 2007).

### Mathematical modeling

The formulation of the models and parameter estimation is described in the Appendix Supplementary Methods. The model was implemented and solved numerically in MATLAB (R2019b).

### Machine learning

We used Python version 3.6.5 and implemented TensorFlow 1.11.0 for neural network design and trainings/validations/tests. During data preprocessing, we used min–max scaling to normalize all the input parameters to be within the region of 0–1. Our neural networks have nine input parameters and one output, which corresponds to the colony biomass. Between the input layer and output layer, we have four fully connected layers with 512, 512, 256, and 128 nodes, respectively. We used the HE initialization method and leaky RELU as activation function with the negative slope α = 0.2 (He *et al*, 2015; Géron, 2017). We chose the initial learning rate to be 0.001, and we performed adaptive moment estimation and gradient clipping to prevent exploding gradients. Moreover, we trained three neural networks and implemented an ensemble method to get the final prediction (Wang *et al*, 2019). This method can further reduce prediction errors and make neural network predictions more accurate and reliable.

## Data availability

The MATLAB codes used for data generation and/or analysis in the study are available on GitHub: https://github.com/youlab/Optimal Patterns_NanLuo.

**Expanded View** for this article is available online.

## Acknowledgements

We thank John Neu, Feilun Wu, Emrah Simsek, Teng Wang, and Anita Silver for discussions and comments. We also thank Caroline Connor for assistance with editing the manuscript. We thank Duke Compute Cluster for assistance with high-throughput computation. We thank Joao Xavier (Memorial Sloan Kettering Cancer Center) for sharing *Pseudomonas aeruginosa* PA14 strains. This study was partially supported by the National Science Foundation (MCB-1937259), the Office of Naval Research (N00014-20-1-2121), and the David and Lucile Packard Foundation.

## Author contributions

NL conceived the research, designed and performed both modeling and experiments, interpreted the results, and wrote the manuscript. SW developed codes for the neural networks, performed data training and predictions using the neural networks, and assisted with manuscript revisions. JL assisted with modeling, experiments, result interpretation, and manuscript revisions. XO assisted with modeling, experiments, and result interpretation. LY conceived the research, assisted in research design, interpreted the results, and wrote the manuscript.

## Conflict of interest

The authors declare that they have no conflict of interest.

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
