## [Review Process File · Molecular Systems Biology]

Collective colony growth is optimized by branching pattern formation in *Pseudomonas aeruginosa*

Nan Luo, Shangying Wang, Jia Lu, Xiaoyi Ouyang, and Lingchong You
DOI: [10.15252/msb.202010089](https://doi.org/10.15252/msb.202010089)

Corresponding author(s): Lingchong You (you@duke.edu)

Review Timeline:	Submission Date:	29th Oct 20
	Editorial Decision:	20th Nov 20
	Revision Received:	9th Feb 21
	Editorial Decision:	9th Mar 21
	Revision Received:	13th Mar 21
	Accepted:	15th Mar 21

Editor: Maria Polychronidou

Transaction Report:

Thank you again for submitting your work to Molecular Systems Biology. We have now heard back from the three referees who agreed to evaluate your study. As you will see below, the reviewers are overall supportive. However, they raise a series of concerns, which we would ask you to address in a major revision.

I think that the recommendations of the referees are rather clear and therefore I see no need to repeat any of the points listed below. Please let me know in case you would like to discuss in further detail any of the issues raised. All issues raised by the referees would need to be satisfactorily addressed.

On a more editorial level, we would ask you to address the following points.

REFeree REPORTS

Reviewer #1:

1. Summary

In this work, the authors developed a new approach to modeling the growths of microbial colonies on surfaces. Specifically, they developed a "zoomed out" model that doesn't consider the microscopic interactions (e.g., movements of individual cells and cell-cell interactions). Instead, the authors first asked what is a "problem" that a microbial colony needs to solve when it expands across a surface: how to colonize as much area as possible given multiple constraints such as

nutrient availability/diffusion, motility of cells, etc.). They then determined the optimal solutions to this problem for several different sets of constraints (i.e., the optimal colony shapes). They then experimentally verified these by growing *P. aeruginosa* strains on agar surfaces with various constraints - examples include agar stiffness (affects cell motility), mutations that affect the cells' swimming ability (affects motility), nutrient availability in the agar, etc. The authors found that, surprisingly, the experimentally observed colonies have nearly the same number of branches, branch widths, and some other key features as the optimal colony shapes that the model predicts. The experimental and model-produced colonies match quite strikingly in their overall shapes, which supports the authors' claim that the real, *P. aeruginosa* colony shapes result from the colonies optimally expanding given the combination of constraints mentioned above.

2. Overall recommendation

Most labs, mine included, study microbial growth in mixing liquid cultures. But real microbes grow on or beneath solid surfaces (e.g., soil or intestines). Relatively few studies have examined microbial colony growths on surfaces, which is more "biologically relevant" than what most of us (my lab) does. This is just one reason that the study is impactful.

Another (perhaps more) important reason that the study is impactful is that this represents a different way of modeling than the usual way of modeling colony growths. As the authors point out in the Discussion section, the typical strategy to model colony growth is to look at cell-cell interactions and focus on cells as individual agents that move ("diffuse"), replicate, and die (the latter two are modelled as "reactions"). The authors' approach is novel: focusing not on the typical microscopic (reaction-diffusion-type) modeling but rather, focus on a zoomed-out view that tells us something new: what is a colony really trying to do to survive and if we understand that, then can we come up with a simpler way of understanding how colonies grow that doesn't rely on the more difficult, microscopic models that often do not give us a "big picture" view of what's going on? The authors succeeded in doing so here. This is fresh approach to modeling that I think will motivate similar approaches to studying not just colony growth but growth of other aggregates of cells (e.g., organoids on dishes).

Third, I was really surprised to see the side-by-side comparison between the model-produced, optimal shapes of colonies and the experimentally observed shapes of colonies, both under a similar set of constraints. The match isn't perfect, since each colony shape is a stochastic outcome (due to stochastic growths and deaths of each cell - more on this below as "Major comment"). I think some controls are missing now but if these colonies weren't hand-picked out of thousands that didn't look like the simulations (I don't think so), then the match between the model and experiment is really impressive and it further strengthens their more "simplified" approach to modeling a complex process (colony growth).

In terms of approach, I like that this manuscript starts with an observation and a model (Fig. 1), develops the model further, and then presents experimental verifications. They did this instead of showing lots of experiments first and then presenting a model (typical for systems biology papers which I'm also guilty of). I think that publishing this work will support Rob Philip's idea of having a model in Fig. 1 instead of in Fig. 7 (i.e., start with a simple idea, make calculations, and then do experiments to check them instead of making an experimental discovery first and then do calculations to explain the discovery). This is often difficult to do and one method isn't necessarily "better" than the other. But the approach of "model/idea first and then check" is rarer than the other approach and it would be a fresh approach (I believe many in systems biology community would agree). The authors have done an admirable job of taking this approach.

Finally, in terms of doors that this paper will open, I believe the work raises interesting questions regarding evolution. Specifically, somehow the colony-shape formation is genetically encoded in individual cells and I suppose that these cells were selected by evolution. But how does a single or even hundreds of cells "know" the macroscopic colony shape that results (i.e., how many branches to form and with what widths, etc.)? The overall colony is shaped by self-organizing forces (nutrient diffusion, motility, etc.) but still, it's an interesting question that I think touches many parts of microbial biology. The general question of how large structures form and how these shapes are encoded in individual cells is an open and interesting question that I think this work motivates one to further pursue in the future.

For the reasons mentioned above, I enthusiastically support the publication of this work in MSB. But I have some major comments regarding control experiments and the model. If the authors can reasonably address these with the COVID constraints in mind, I think this interesting work can reach a wider audience and be backed by more solid arguments than it is now.

3. Major comments

Major point 1:

What actually causes the branches to form, how many of them to form, and their widths?

The model, at least in the main text, doesn't really tell us this. The main text simply says that the model predicts these quantities and that you observe them in the selected images from experiments (more on this below). I read the supplements and followed the calculations. While I can kind of see how these arise from these calculations, these calculations aren't that simple. I think it's important to explain the mechanics of the model in words in the main text. The number of branches, their widths, and the densities of branches per length all result from a "self-organization" process (i.e., due to the constraints that the authors looked at such as agar density and nutrient availability). But it's important to describe these in words and in the main text so that a reader who doesn't read the supplement to get the main ideas behind the model. I'd also mention that this - the self-organization process - is probably also why one doesn't need to look at microscopic interactions as in reaction-diffusion-type equations.

In general, I thought that one of the main weaknesses of this manuscript, as it's formatted now, is that one cannot really understand the model just from reading the main text because very few details about the model are really given after Figure 1. Between Fig 1 and Fig 2, I'd just explain in words what constraints cause a branch to appear (I think this is the most important feature of the model to understand) and also explain what things determine where a branch appears and how many of them to appear per length of another branch.

Major point 2:

While it's nice to see colony images from both experiments and simulations matching so nicely, I couldn't clearly see how many colonies were analyzed per experimental condition. Most of the figures show one or few colony images from experiments and then similar looking colonies from the model next to these images. But the authors should really quantify how often they see a match and how "good" the match is, when they use the same condition to grow multiple colonies (each on a different plate). As the authors know, the colony shape will vary from plate to plate, due to stochastic growths, deaths, and movements of cells. But will the macroscopic features like the # of branches, their widths, and branch densities survive these stochastic variations? I couldn't see this

addressed for all/most of the experimental conditions. I suggest making a histogram that shows the number of colonies that exhibit each macroscopic feature. This is an important control and it will also reveal how "accurately" cells optimally solve the problem of colony expansion. The authors don't need to do this for every experimental condition but for some of the key ones, I think they need to do some of these statistics / controls (at least 10 plates and less than 100 plates).

Major point 3:

Related to major point 1. The text doesn't give much intuition for some of the main results. For example, the main text doesn't intuitively explain, through the model, why a lower agar density leads to non-branching colonies to expand and grow more efficiently than the branching ones (lines 143-145). Give some intuition and at least explain how the model comes to this conclusion. Right now, the main text is simply reporting on what the model in the supplement is predicting, without explaining what the model is and how it comes to these conclusions.

- Another example of a model prediction (and experimental verification) that doesn't explain why the phenomenon occurs: "Consistent with the predictions, the observed patterns of wild-type *Pseudomonas* colonies under ... branches formed by *Pseudomonas* colonies become wider and denser (Figure 3B)" (lines 149-152). The main text is scattered with statements like this. Please give an intuitive explanation for every such statement.

Minor comments:

1. The authors used PA14 strain and its mutants obtained from experimental evolution (hyperswarming mutants with the flagellar synthesis regulator gene). In the main text, explain how these strains were selected (i.e., the actual evolution experiment).

2. How do you define a growth domain? For a single branch, is it the total area (= width X (curved) length)? This isn't clearly specified.

3. Why is $WD \geq 1$ for a colony without any branches (so a regular, circular colony has $WD \geq 1$)? Explain in words.

4. The authors define the expansion efficiency (γ) only in the equation that's embedded in Fig. 1. But for the main text's discussion of Fig. 2A, it's important to mention what γ represents in words. One can only understand what γ is much later in the text, when the authors change the agar stiffness.

5. Scale bars are missing for all pictures of colonies. No need to show a scale bar for every picture. Just show one in Fig. 1A so that a picture shows how wide a plate is.

6. In Fig. 1C: Define nutrient N and cell density C like you do for W , D , L , and R above the equations.

7. Eliminate the "only" in "The growth rate of hyperswarmers in liquid cultures is only 7% lower than that of wild-type" (bottom of Pg. 6). 7% is actually large, given exponential growth, no? I also don't see why this sentence is necessary here.

Reviewer #2:

In this paper, authors applied the optimality paradigm to understanding and predicting dendrite pattern formation in growing colonies of bacteria. Instead of analyzing detailed reaction-diffusion models for bacterial growth, motility and nutrient consumption that have been the staple of research in this topic in the last 25-30 years, they took a different approach. They postulated at first a very simple phenomenological model where the branches have fixed width W and density D per unit area. Cells that comprise these branches consume nutrient in proportion to their growth and extend in the direction of the nutrient gradient. The rate of expansion of the colony depends on the choice of parameters W and D (along with other parameters of the model) in a non-trivial way. By systematically varying W and D authors obtained "fitness landscapes", where the fitness is defined as the growth rate of the total cell mass. For nutrient-rich and rapidly extending systems, optimal growth is achieved near the line $WD=1$, which corresponds to very thick branches or no branching at all. On the contrary, in nutrient-poor case or very slow branch extension, thin and sparse branches ($WD \ll 1$) yield the maximum cell mass growth. Authors took this observation to postulate that during colony growth, the optimum combination of W and D is selected locally depending on the nutrient availability, and so this local optimality can be used to predict the pattern structure. Indeed, comparing this modeling approach with their own experiments with *Pseudomonas aeruginosa* strain of bacteria, they found a pretty convincing (however qualitative) evidence that patterns follow this general optimality principle. Another interesting aspect of their modeling approach is that they used machine learning to accelerate the exploration of the parameter space that otherwise would involve many time-consuming runs of the model. This approach can potentially be useful in a variety of situations where large parameter space needs to be explored.

I found the paper interesting and thought provoking. I think it can eventually be published, however there are a number of issues that need to be addressed first. The biggest drawback in the presented results, in my opinion, is the lack of quantitative analysis of experimentally observed patterns in terms of the width and density of branches. It seems that authors can easily control the density of branches in 1D experiments by varying the distance between inoculated bacterial patches. If so, they could measure the rate of extension, width of resulting branches, and the cell growth rate to demonstrate the system indeed has an optimal W for a given D . If these additional experiments are done, the comparison between the model and experiments can be quantitative rather than qualitative.

Another point that was not clear to me is the comparison of the wild type *Pseudomonas* with a "hyperswarmer" mutant strain. It seems that the authors assumed that hyperswarmers simply have $WD=1$ and therefore do not form branches but otherwise have the same parameters as the wild-type. However, the apparent lack of branching phenotype in hyperswarmers under normal conditions is not a feature that is controlled by an independent parameter, but rather a consequence of certain phenotypical changes on a single-cell level. In particular, according to (van Dimarsch et al, 2013), hyperswarmers have significantly higher swarming motility (hence the name) and so, presumably, would have a higher parameter γ for the same agar concentration. So, effectively, hyperswarmers would automatically belong to the "High expansion efficiency" regime (Fig, 2A,c). Furthermore, van Dimarsch et al. also note that due to higher chemotactic motility hyperswarmers tend to concentrate near the colony front, and that gives them a competitive advantage over wild type in swarming conditions. But if this is the case, the simple model of uniform cell distribution within the colony may not be appropriate for the hyperswarmers. Thus, the comparison of hyperswarmers and the wild type for validating the optimization approach may not

be entirely justified.

Among minor issues, I will note the following:

1. In the 2D extension of the model, authors write that branches are split to maintain approximately constant local density D which is defined as the inverse distance between two neighboring branch tips. However, when a branch bifurcates, the distance between the two tips is very small, which would correspond to a very high D . More details should be presented to how precisely their algorithm works.
2. Supp I.2: What exactly are the initial conditions: is it a circle, or there are some nascent branches in the beginning to stipulate the number of initial branches?
3. Supp I.2: How the area of a branch is defined if the branch splits?
4. In Fig. 1C and in Supplemental Eq.(3), authors used a partial time derivative of C , however since C is the total cell mass which is only a function of time, the derivative should be ordinary.
5. There is an incorrect spelling in the citation (Jacob 2004) In fact, the first author is E. Ben-Jacob.

Reviewer #3:

Theoretical approaches to understanding biological pattern formation typically attempt to explain the bottom-up, mechanistic processes that lead to patterns. This study takes an opposing, top-down approach, finding the patterns that would theoretically optimise growth given a set of environmental conditions and comparing them to those observed experimentally under those conditions. The specific system used to test this conceptual framework is the *P. aeruginosa* swarming motility system, which generates branching patterns as cells move over an agar surface. Using a simple, nutrient-based model of colony growth, the authors predict the yield of colonies with differing branching behaviours and find the geometrical parameters of the branching network (branch density and width) that optimise this yield for a given set of environmental conditions. Experiments suggest that the patterns formed by real colonies in variable environments follow similar trends to the theoretical optimum.

The theory behind this study is elegant, providing an important insight into the ecological constraints on branching pattern formation using a framework that should easily generalize to other systems. Because branching is common to a wide variety of systems other than the *P. aeruginosa* swarming assay used in this study, the results are likely to be of broad interest to a variety of experimental biologists, as well as biophysicists and mathematical biologists. However, I think that the data from the illustrative example of the swarming assay needs to be more carefully analysed to properly support their conclusions.

Major comments:

- It should be made much clearer throughout the text that the experimental system is based on the swarming assay. Without this context, statements such as 'Typically, *Pseudomonas* colonies form branches that extends in two dimensions (2D) when initiated from a point inoculation' (line 93) are not true - the branching effect observed in swarming colonies is not typical of many surface-grown *Pseudomonas* colonies, but is observed under a fairly narrow range of experimental conditions (Tremblay, J. and Déziel, E., 2008).

- It is hard to tell by eye whether the width of the branches of the colonies in Figs. 2B and 3C is following the predicted trends. The authors' case would be considerably strengthened if they could measure this across all their replicates and present this data alongside the representative images.

Quantifying the branch density of the colonies would also be useful.

- *P. aeruginosa* can regulate swarming motility according to environmental conditions, changing it based on e.g. the main carbon source available (Shrout et al. 2006), the availability of nitrate (Déziel, E. et al. 2003), or the agar concentration (Kamatkar and Shrout, 2011). However, the modelling approach assumes that the expansion rate of the branches (a proxy for cell motility) is fixed, rather than a physiological parameter that can itself be optimized. While I can appreciate the elegance of restricting the optimisation to the geometrical properties of the branches, the possibility that the trade-off between expansion speed and growth rate may also be a target of evolutionary optimization should at least be mentioned in either the introduction or the discussion.

- Can the use of the cell-density dependent terms (i.e. $\frac{cmC}{(cm + C)}$) in Eq. 2 of the supplement be more thoroughly justified? As far as I can tell, this expression is purely based on the early growth rate data in Supplementary Fig. 1A, which shows slower growth between $t = 0-4$ h. However, this is likely to simply be the lag phase following transferral of cells from stationary-phase culture to fresh media (lines 364-365), rather than a cell density-dependent effect. I would expect that preparing the growth curve plate from an exponential-phase sub-culture rather than the overnight culture would remove this early stage lag.

If these terms are not necessary, I recommend that they be removed for simplicity. The expression for f_G will then simplify to the Monod equation, the standard model for the relationship between nutrient concentration and growth rate (e.g. Shluter et al. 2015, Reino et al. 2016).

Minor comments:

Figures:

- A scalebar should be added to at least one of the images in each set of colony images.
- Fig. 2A - colourbars need labels (e.g. 'Biomass', as in Fig. 3)
- Fig. 2B - are all the images taken at the same time following inoculation? I am confused by sub-panel (c), which seems to show a colony that has expanded less far than the colony in (a) despite apparently being grown under conditions that permit faster expansion. Please make the imaging times clear either in the caption or as part of the figure.
- Supplementary Fig. 4, condition $D=0.15$ $W=1.5$ - This particular simulation seems to have been buggy, as there are large gaps in the colony perimeter that should have been filled in by the branching algorithm.
- Supplementary Fig. 5 - needs colourbar

Text:

- I can't find experimental details of how Supplementary Figure 3 was prepared. How does 'solid LB media' differ from the standard swarming media with equal agar concentration?
- Lines 143 - 146: Figure callouts of Fig. 2B need to specify the sub-panel referred to.
- In general, more care should be taken to proof-read the manuscript - e.g. edit line 39-40 to '...which are dwarfed in complexity and sophistication by patterns found in nature.', line 239 to '...more sophisticated descriptions of cell motility have been used...

Point-by-point responses to reviewers' comments

Reviewer #1:

1. Summary

In this work, the authors developed a new approach to modeling the growths of microbial colonies on surfaces. Specifically, they developed a "zoomed out" model that doesn't consider the microscopic interactions (e.g., movements of individual cells and cell-cell interactions). Instead, the authors first asked what is a "problem" that a microbial colony needs to solve when it expands across a surface: how to colonize as much area as possible given multiple constraints such as nutrient availability/diffusion, motility of cells, etc.). They then determined the optimal solutions to this problem for several different sets of constraints (i.e., the optimal colony shapes). They then experimentally verified these by growing *P. aeruginosa* strains on agar surfaces with various constraints - examples include agar stiffness (affects cell motility), mutations that affect the cells' swimming ability (affects motility), nutrient availability in the agar, etc. The authors found that, surprisingly, the experimentally observed colonies have nearly the same number of branches, branch widths, and some other key features as the optimal colony shapes that the model predicts. The experimental and model-produced colonies match quite strikingly in their overall shapes, which supports the authors' claim that the real, *P. aeruginosa* colony shapes result from the colonies optimally expanding given the combination of constraints mentioned above.

2. Overall recommendation

Most labs, mine included, study microbial growth in mixing liquid cultures. But real microbes grow on or beneath solid surfaces (e.g., soil or intestines). Relatively few studies have examined microbial colony growths on surfaces, which is more "biologically relevant" than what most of us (my lab) does. This is just one reason that the study is impactful.

Another (perhaps more) important reason that the study is impactful is that this represents a different way of modeling than the usual way of modeling colony growths. As the authors point out in the Discussion section, the typical strategy to model colony growth is to look at cell-cell interactions and focus on cells as individual agents that move ("diffuse"), replicate, and die (the latter two are modelled as "reactions"). The authors' approach is novel: focusing not on the typical microscopic (reaction-diffusion-type) modeling but rather, focus on a zoomed-out view that tells us something new: what is a colony really trying to do to survive and if we understand that, then can we come up with a simpler way of understanding how colonies grow that doesn't rely on the more difficult, microscopic models that often do not give us a "big picture" view of what's going on? The authors succeeded in doing so here. This is fresh approach to modeling that I think will motivate similar approaches to studying not just colony growth but growth of other aggregates of cells (e.g., organoids on dishes).

Third, I was really surprised to see the side-by-side comparison between the model-produced, optimal shapes of colonies and the experimentally observed shapes of colonies, both under a similar set of constraints. The match isn't perfect, since each colony shape is a stochastic outcome (due to stochastic growths and deaths of each cell - more on this below as "Major comment"). I think some controls are missing now but if these colonies weren't hand-picked out of thousands that didn't look like the simulations (I don't think so), then the match between the model and experiment is really impressive and it further strengthens their more "simplified" approach to modeling a complex process (colony growth).

In terms of approach, I like that this manuscript starts with an observation and a model (Fig. 1), develops the model further, and then presents experimental verifications. They did this instead of showing lots of experiments first and then presenting a model (typical for systems biology papers which

I'm also guilty of). I think that publishing this work will support Rob Philip's idea of having a model in Fig. 1 instead of in Fig. 7 (i.e., start with a simple idea, make calculations, and then do experiments to check them instead of making an experimental discovery first and then do calculations to explain the discovery). This is often difficult to do and one method isn't necessarily "better" than the other. But the approach of "model/idea first and then check" is rarer than the other approach and it would be a fresh approach (I believe many in systems biology community would agree). The authors have done an admirable job of taking this approach.

Finally, in terms of doors that this paper will open, I believe the work raises interesting questions regarding evolution. Specifically, somehow the colony-shape formation is genetically encoded in individual cells and I suppose that these cells were selected by evolution. But how does a single or even hundreds of cells "know" the macroscopic colony shape that results (i.e., how many branches to form and with what widths, etc.)? The overall colony is shaped by self-organizing forces (nutrient diffusion, motility, etc.) but still, it's an interesting question that I think touches many parts of microbial biology. The general question of how large structures form and how these shapes are encoded in individual cells is an open and interesting question that I think this work motivates one to further pursue in the future.

For the reasons mentioned above, I enthusiastically support the publication of this work in MSB. But I have some major comments regarding control experiments and the model. If the authors can reasonably address these with the COVID constraints in mind, I think this interesting work can reach a wider audience and be backed by more solid arguments than it is now.

We thank the reviewer for recognizing the conceptual novelty and significance of the work. We also thank the reviewer for the insightful comments and suggestions.

3. Major comments

Major point 1:

What actually causes the branches to form, how many of them to form, and their widths?

The model, at least in the main text, doesn't really tell us this. The main text simply says that the model predicts these quantities and that you observe them in the selected images from experiments (more on this below). I read the supplements and followed the calculations. While I can kind of see how these arise from these calculations, these calculations aren't that simple. I think it's important to explain the mechanics of the model in words in the main text. The number of branches, their widths, and the densities of branches per length all result from a "self-organization" process (i.e., due to the constraints that the authors looked at such as agar density and nutrient availability). But it's important to describe these in words and in the main text so that a reader who doesn't read the supplement to get the main ideas behind the model. I'd also mention that this - the self-organization process - is probably also why one doesn't need to look at microscopic interactions as in reaction-diffusion-type equations.

In general, I thought that one of the main weaknesses of this manuscript, as it's formatted now, is that one cannot really understand the model just from reading the main text because very few details about the model are really given after Figure 1. Between Fig 1 and Fig 2, I'd just explain in words what constraints cause a branch to appear (I think this is the most important feature of the model to understand) and also explain what things determine where a branch appears and how many of them to appear per length of another branch.

We thank the reviewer for pointing out the ambiguity in our text. We edited the main text to clarify how exactly our model makes the predictions. Specifically, in the Results, we expanded the first section to explain the details of our model formulation and the first two paragraphs of the third section to elaborate on how branches initiate and bifurcate in our model.

A major difference that distinguishes our model from other biophysical models is that it does not investigate the molecular mechanisms of branch formation. We focus not on what causes the formation of branches, but on the *consequences* of branches that are formed, i.e., how the combination of branch width and density affects the efficiency of biomass accumulation. To this end, we built an optimization model, in which we calculate the biomass accumulation of colonies with different branch widths and densities to find the optimal configurations under a given growth condition. With these data, we further developed a model that can predict colony patterns under more complex conditions. In this model, the initiation of branches is artificially imposed, but the density and width of branches are dictated by the optimal numbers we obtained from optimization modeling.

Following are our point-by-point responses to the specific questions raised by the reviewer:

- “*What actually causes the branches to form*” – Our model is agnostic about how the branches emerge but rather focuses on the consequences of branching. In typical biophysical models, the branches emerge from dynamical instability at the interface between the colony and the growth substrate (Ben-Jacob *et al.*, 1994, Farrell *et al.*, 2013, Giverso *et al.*, 2015, Kawasaki *et al.*, 1997, Kozlovsky *et al.*, 1999, Matsushita *et al.*, 1998, Mimura *et al.*, 2000, Trinschek *et al.*, 2018).
- “*How many of them to form*”, “*The number of branches, their widths, and the densities of branches per length*” – The density and width of branches developed under a particular local environment are determined by optimization modeling, which gives us a mapping between the optimal branch characteristics and the growth conditions (as shown in Figure 3A).
- “*What constraints cause a branch to appear*”, “*What things determine where a branch appears and how many of them to appear per length of another branch*” – When we have the local density of branches (D), we can predict where subbranches appear: Specifically, we track the tip of a growing branch and calculate the actual local branch density; as the branch extends, if the actual local branch density becomes lower than the predicted density, the code triggers branch bifurcation and initiate a new subbranch.

Major point 2:

While it's nice to see colony images from both experiments and simulations matching so nicely, I couldn't clearly see how many colonies were analyzed per experimental condition. Most of the figures show one or few colony images from experiments and then similar looking colonies from the model next to these images. But the authors should really quantify how often they see a match and how "good" the match is, when they use the same condition to grow multiple colonies (each on a different plate). As the authors know, the colony shape will vary from plate to plate, due to stochastic growths, deaths, and movements of cells. But will the macroscopic features like the # of branches, their widths, and branch densities survive these stochastic variations? I couldn't see this addressed for all/most of the experimental conditions. I suggest making a histogram that shows the number of colonies that exhibit each macroscopic feature. This is an important control and it will also reveal how "accurately" cells optimally solve the problem of colony expansion. The authors don't need to do this for every experimental condition but for some of the key ones, I think they need to do some of these statistics / controls (at least 10 plates and less than 100 plates).

This is a great suggestion and we agree that a quantified comparison between simulations and images will strengthen our arguments. We provided details of statistics in the legends of each figure.

Using an optimization model, we showed that the optimal patterns of colonies tend to have thin branches under starvation or when cells have low mobility (Figure 3A). We showed the images of colonies growing under different conditions to demonstrate that *Pseudomonas* colonies on swarming media generally follow this rule (Figure 3B).

As suggested by the reviewer, we quantified the branch width of the images. Figure R1 shows the predicted optimal branch width (panel A) and the average branch width of *Pseudomonas* (panel B), and they follow a similar trend: the branch width increases with increasing nutrient concentration or decreasing agar density in the medium. For statistical analysis, we pooled samples grew under similar conditions together (panel C) and calculated the average branch width of four condition groups (panel D), which shows a significant increase in branch width in colonies on rich media or media with low agar densities. In both the model and experimental data, the relationship between the branch density and the nutrient concentration or the expansion efficiency varies and depends on other parameters.

Figure R1. Branch width of *Pseudomonas* colonies increases with increasing nutrient concentration and decreasing agar density. **A.** The optimal branch widths (mm; shown by numbers and colors in the table) predicted by the optimization model with different combinations of environmental parameters. **B.** The average branch widths (mm; shown by numbers and colors in the table) of *Pseudomonas* colonies under different combinations of growth conditions. The mean branch width of a colony was measured in a semi-automated manner (described in detail in Methods and Protocols). For each condition, the average of the mean branch widths of 2-4 colonies was shown. **C.** The growth conditions of *Pseudomonas* colonies are divided into four groups: I. Low nutrient concentration (4 g/L – 8 g/L casamino acids) and high agar density

(0.50%-0.55%); II. High nutrient concentration (10 g/L – 16 g/L casamino acids) and high agar density (0.50%-0.55%); III. Low nutrient concentration (4 g/L – 8 g/L casamino acids) and low agar density (0.40%-0.45%); IV. High nutrient concentration (10 g/L – 16 g/L casamino acids) and low agar density (0.40%-0.45%). **D.** The average branch width of each group in **C**. The sample size is the number of colonies being measured in each group. Error bars show standard deviations. *** $P < 0.001$, **** $P < 0.0001$ (Student's t test).

To demonstrate the predictive power of our model, we simulated colony patterns in three scenarios (Figures 4, 5, and 6) and compared the simulations to experimental images. The consistency in the branch width between the simulations and images is demonstrated above (Figure R1). In the revision, Figure R1 is presented as Figure EV2.

Major point 3:

Related to major point 1. The text doesn't give much intuition for some of the main results. For example, the main text doesn't intuitively explain, through the model, why a lower agar density leads to non-branching colonies to expand and grow more efficiently than the branching ones (lines 143-145). Give some intuition and at least explain how the model comes to this conclusion. Right now, the main text is simply reporting on what the model in the supplement is predicting, without explaining what the model is and how it comes to these conclusions.

Another example of a model prediction (and experimental verification) that doesn't explain why the phenomenon occurs: "Consistent with the predictions, the observed patterns of wild-type *Pseudomonas* colonies under branches formed by *Pseudomonas* colonies become wider and denser (Figure 3B)" (lines 149-152). The main text is scattered with statements like this. Please give an intuitive explanation for every such statement.

We thank the reviewer for pointing this out. We analyzed the spatial-temporal dynamics of the system (Figure EV1) and gave an intuitive explanation of why the model predicts different optimal patterns under different conditions. Basically, the optimal patterns depend on the spatial distribution of accessible nutrient: With limited nutrient or a high agar density, nutrient is quickly depleted in the center of the colony, so the utilization of nutrient mainly occurs at the colony front (Figure EV1A). Therefore, the amount of nutrient utilization is correlated with the length of the colony boundaries, which is greater in colonies with thin branches. With rich nutrient or a low agar density, however, colonies expand before consuming all the nutrient in the area covered by cells (Figure EV1B,C). Hence, the consumption of nutrient is also related to the colony area, which is higher in non-branching colonies that expand uniformly with no gaps.

Minor comments:

1. The authors used PA14 strain and its mutants obtained from experimental evolution (hyperswarming mutants with the flagellar synthesis regulator gene). In the main text, explain how these strains were selected (i.e., the actual evolution experiment).

We apologize for not explaining this clearly in the main text. We added a paragraph describing how this evolution experiment was performed:

“We obtained the mutants using experimental evolution as described in van Ditmarsch *et al.*, 2013. *Pseudomonas aeruginosa* PA14 was grown on swarming media for 20 hours and the entire colony was collected from the plate. A fraction of the collected cells was inoculated on a new

plate with swarming media. We repeated this procedure for seven consecutive days and isolated mutants that do not develop branching patterns but formed irregular or circular colonies, a phenotype called hyperswarming (van Ditmarsch *et al.*, 2013).”

2. How do you define a growth domain? For a single branch, is it the total area (= width X (curved) length)? This isn't clearly specified.

We define the growth domain of a colony as the plate it grows in; or in simulations, the spatial domain constraining the differential equations. When the colony develops branches, the growth domain is partitioned into smaller domains for each branch. In the 1D scenario where the branches are parallel and evenly distributed, the growth domain is equally partitioned and the area of each smaller domain is A/n (A = area of the entire growth domain; n = number of branches). Since this phrase is indeed likely to be confusing, we replaced “the growth domain” in the main text by “the plate”, “the growth medium”, or “the rectangular domain” depending on the context.

3. Why is $WD \geq 1$ for a colony without any branches (so a regular, circular colony has $WD \geq 1$)?. Explain in words.

The branch density, D , is defined as the number of branches per unit length. Therefore, the total number of branches in a domain with a size of R is DR . Since each branch has a width of W , the total width of all branches is WDR . When $WDR \geq R$ (i.e., $WD \geq 1$), the total width of branches exceeds the domain size, so branches fuse together or overlap with each other and the colony becomes non-branching or circular.

Another way to interpret this is to consider the distance between branches: when the branch density is D , the distance between the midlines of two branches is $1/D$. If $W \geq 1/D$ (i.e., $WD \geq 1$), two branches overlap and the colony looks circular.

We thank the reviewer for asking this question and added the explanation to the main text.

4. The authors define the expansion efficiency (γ) only in the equation that's embedded in Fig. 1. But for the main text's discussion of Fig. 2A, it's important to mention what γ represents in words. One can only understand what γ is much later in the text, when the authors change the agar stiffness.

Thanks for pointing this out. In the revised manuscript, we elaborated on the physical meaning of γ before we discuss Figure 2A:

“Here, γ is a coefficient relating the amount of energy for cell movement to the expansion rate of the colony. Using the same amount of energy, with greater γ , the colony expands faster, so we hitherto refer to γ as the colony expansion efficiency. We can experimentally change the expansion efficiency by altering the agar density of the swarming media, as *Pseudomonas* colonies expand faster on media with lower agar densities (Appendix Figure S2).”

5. Scale bars are missing for all pictures of colonies. No need to show a scale bar for every picture. Just show one in Fig. 1A so that a picture shows how wide a plate is.

Thanks for pointing this out. We added scale bars to at least one image in each figure.

6. In Fig. 1C: Define nutrient N and cell density C like you do for W, D, L, and R above the equations.

We added the definition of N and C above the equations in Figure 1C.

7. Eliminate the “only” in “The growth rate of hyperswarmers in liquid cultures is only 7% lower than that of wild-type” (bottom of Pg. 6). 7% is actually large, given exponential growth, no? I also don’t see why this sentence is necessary here.

We made this comparison to exclude the possibility that the observed disadvantage of hyperswarmers under resource-limited conditions (Figure 2B panel a) may be due to the reduction in growth rate. As we show in Supplementary Figure 1B, hyperswarmers only have a small growth disadvantage when growing in liquid cultures (leading to an average of 10.7% reduction in biomass accumulation when growing with 4 g/L casamino acids). However, when growing on solid media with the same concentration of nutrient (Figure 2B, panel a), hyperswarmer colonies grow much worse than wild-type (46%-79% reduction in colony areas), which cannot be explained by the lower growth rate of hyperswarmers.

We thank the reviewer for pointing out the lack of clarity in delivering this message. We have revised the main text to better convey the points above.

Reviewer #2:

In this paper, authors applied the optimality paradigm to understanding and predicting dendrite pattern formation in growing colonies of bacteria. Instead of analyzing detailed reaction-diffusion models for bacterial growth, motility and nutrient consumption that have been the staple of research in this topic in the last 25-30 years, they took a different approach. They postulated at first a very simple phenomenological model where the branches have fixed width W and density D per unit area. Cells that comprise these branches consume nutrient in proportion to their growth and extend in the direction of the nutrient gradient. The rate of expansion of the colony depends on the choice of parameters W and D (along with other parameters of the model) in a non-trivial way. By systematically varying W and D authors obtained “fitness landscapes”, where the fitness is defined as the growth rate of the total cell mass. For nutrient-rich and rapidly extending systems, optimal growth is achieved near the line $WD=1$, which corresponds to very thick branches or no branching at all. On the contrary, in nutrient-poor case or very slow branch extension, thin and sparse branches ($WD \ll 1$) yield the maximum cell mass growth. Authors took this observation to postulate that during colony growth, the optimum combination of W and D is selected locally depending on the nutrient availability, and so this local optimality can be used to predict the pattern structure. Indeed, comparing this modeling approach with their own experiments with *Pseudomonas aeruginosa* strain of bacteria, they found a pretty convincing (however qualitative) evidence that patterns follow this general optimality principle. Another interesting aspect of their modeling approach is that they used machine learning to accelerate the exploration of the parameter space that otherwise would involve many time-consuming runs of the model. This approach can potentially be useful in a variety of situations where large parameter space needs to be explored.

I found the paper interesting and thought provoking. I think it can eventually be published, however there are a number of issues that need to be addressed first. The biggest drawback in the presented results, in my opinion, is the lack of quantitative analysis of experimentally observed patterns in terms of the width and density of branches. It seems that authors can easily control the density of branches in 1D experiments by varying the distance between inoculated bacterial patches. If so, they could measure the rate of extension, width of resulting branches, and the cell growth rate to demonstrate the system indeed has an optimal W for a given D . If these additional experiments are done, the comparison between the model and experiments can be quantitative rather than qualitative.

We thank the reviewer for the thorough evaluation of our work and for recognizing and confirming the message we try to deliver. We agree with the reviewer that quantitative analysis will make our work much more solid. Other reviewers raised the same point. Using an optimization model, we show that the optimal patterns of colonies tend to have thin branches under starvation or when cells have low mobility (Figure 3A). We showed the images of colonies growing under different conditions to demonstrate that *Pseudomonas* colonies on swarming media generally follow this rule (Figure 3B).

As the reviewer suggested, we quantified the branch width of the images to support this conclusion (Figure R1), which is presented as Figure EV2 in the revised manuscript. Figure R1 shows the predicted optimal branch width (panel A) and the average branch width of *Pseudomonas* (panel B), and they follow a similar trend: the branch width increases with increasing nutrient concentration or decreasing agar density in the medium. For statistical analysis, we pooled samples grew under similar conditions together (panel C) and calculated the average branch width of four condition groups (panel D), which shows a significant increase in branch width in colonies on rich media or media with low agar densities. In both the model and experimental data, the relationship between the branch density and the nutrient concentration or the expansion efficiency varies and depends on other parameters.

The experiment brought up by the reviewer is a great suggestion. It will be a great validation of our model if we can arbitrarily control the branches of a colony. However, the density and width of branches developed by bacterial colonies of a particular genotype under particular conditions are not easily controllable by controlling the seeding configuration. This is the rationale of how we predict the branching patterns growing under different conditions. Therefore, we tested our model by comparing different bacterial strains or colonies growing under different conditions.

Another point that was not clear to me is the comparison of the wild type *Pseudomonas* with a "hyperswarmer" mutant strain. It seems that the authors assumed that hyperswarmers simply have $WD=1$ and therefore do not form branches but otherwise have the same parameters as the wild-type. However, the apparent lack of branching phenotype in hyperswarmers under normal conditions is not a feature that is controlled by an independent parameter, but rather a consequence of certain phenotypical changes on a single-cell level. In particular, according to (van Dimarsch et al, 2013), hyperswarmers have significantly higher swarming motility (hence the name) and so, presumably, would have a higher parameter γ for the same agar concentration. So, effectively, hyperswarmers would automatically belong to the "High expansion efficiency" regime (Fig. 2A,c). Furthermore, van Dimarsch et al. also note that due to higher chemotactic motility hyperswarmers tend to concentrate near the colony front, and that gives them a competitive advantage over wild type in swarming conditions. But if this is the case, the simple model of uniform cell distribution within the colony may not be appropriate for the hyperswarmers. Thus, the comparison of hyperswarmers and the wild type for validating the optimization approach may not be entirely justified.

This is an excellent point and we agree with the reviewer on the caveat of using mutants to validate our modeling results. In fact, unlike in simulations, it is nearly impossible to alter one parameter (e.g., the pattern) without changing other parameters in a real biological system, because the colony pattern is not an independent variable, but a phenotype emerges from complex interactions of other system variables. Hyperswarmers not only have higher swarming motility, but also produce less surfactant and grow slower than wild-type (van Dimarsch et al, 2013). What we show here is that the pattern of this mutant strain can serve as a coarse-grained predictor of the population fitness. The other cell-level parameters, such as the higher swarming motility of the hyperswarmers, can only explain why hyperswarmer colonies grow faster under some conditions (i.e., high nutrient concentrations or low agar densities, Figure 2B, b and c), but not others (Figure 2B, a). However, we found that the colony pattern is a good indicator of colony fitness under different conditions.

We also used modeling to analyze how the growth of wild-type compared to hyperswarmers if the enhanced motility of hyperswarmers is considered. With a higher cell motility, the actual colony expansion efficiency increased under the same growth conditions (e.g., on media with the same agar density). Figure R2 shows the difference in biomass accumulation between a branching colony ("B", i.e., the wild-type) and a non-branching colony ("NB", i.e., the hyperswarmers). In panel A, we assume both colonies have equal cell motilities. In panel B, we assume the non-branching colony has a 2-fold increase in cell motility (van Dimarsch et al., 2013), so its expansion efficiency doubles. Adjusting the actual expansion efficiency did not change the trends of the data and the conclusions (Figure R2): when nutrient is limited or expansion is restricted by external conditions, branching colonies outperform non-branching ones (yellow regions); with either high nutrient concentrations or high expansion efficiencies, the non-branching colonies grow better (green regions).

Figure R2. Comparison between branching and non-branching colonies considering the difference in cell motility. Colors represent the difference in biomass accumulation between a branching colony (“B”, $W = 2$, $D = 0.1$) and a non-branching one (“NB”, $W = 10$, $D = 0.1$) with varying initial nutrient concentration (horizontal direction) and colony expansion efficiency (vertical direction). **A.** Assuming both colonies have equal cell motilities. **B.** Assuming the non-branching colony has a 2-fold increase in cell motility. As a result, under the same external conditions, the colony expansion efficiency of the non-branching colony is twice that of the branching colony.

The reviewer is also correct in that hyperswarmers have a significant competitive advantage because they concentrate near the colony front. Both van Dimarsch et al. and we observed striking spatial segregation of these two species when they coexist within one colony. However, this segregation is primarily due to competition in a mixture. In a clonal population, we do not observe a significant difference in the spatial distribution of biomass; that is, cells of hyperswarmer colonies do not tend to accumulate more on the colony front than those of wild-type colonies (Figure 2B). Therefore, we did not consider the impact of cell motility on the spatial distribution of cells within a population.

Among minor issues, I will note the following:

1. In the 2D extension of the model, authors write that branches are split to maintain approximately constant local density D which is defined as the inverse distance between two neighboring branch tips. However, when a branch bifurcates, the distance between the two tips is very small, which would correspond to a very high D . More details should be presented to how precisely their algorithm works.

The reviewer is correct in that, in the 2D extension formulation of the model, the local branch density will vary continuously as the colony expands from the center. In fact, this is part of the reason why we chose to start the analysis with a simplified 1D extension formulation.

In the 2D simulation, we apply a simple rule to convert the continuous variable (branch density) to discrete bifurcation points (Figure R3). A branch bifurcates when the local branch density is sufficiently low (below the average, optimal density D). Upon branching, the local branch density will double but it will gradually decrease until the next branching event. We choose the threshold branch density for triggering bifurcation so that the branch density will oscillate around D . Figure R3 shows an example: Panel A shows the actual pattern; as the colony grows, the actual distances between branch tips (colored curves in panel B) fluctuate around the inverse of D (the black line in panel B).

Figure R3. Temporal dynamics of branch densities in a simulation of a colony expanding in 2D. **A.** The simulated pattern of a colony expanding in 2D. **B.** The local inter-branch distance of each branch tip. Each colored curve represents one branch in A. The black line indicates the inverse of the given branch density, D . Each time a branch bifurcates, the distance between tips drops below $1/D$; as the two new tips separate, their distance gradually increases over $1/D$.

2.Supp I.2: What exactly are the initial conditions: is it a circle, or there are some nascent branches in the beginning to stipulate the number of initial branches?

We apologize for the ambiguity. The initial cell density is $C = a \text{ constant}$ within the colony. The initial shape of the colony is a circle. When we are simulating the branch development, we are simulating how each branch tip move (branch extension) and split (branch bifurcation). Hence, another important initial condition we missed is the initial number of branch tips, which is equal to $2\pi r_0 D$ (the circumference of the circular colony times the branch density). We included this information in the revised supplementary materials.

3.Supp I.2: How the area of a branch is defined if the branch splits?

This is a great question. We calculate the area of the i -th branch, A_i , with

$$A_i = \int_{\Omega_i} \frac{1}{n} dx,$$

where Ω_i is the region of the i -th branch and n is the number of branches sharing the same area.

Figure R4 illustrates how we calculate the area of a branch in the 2D extension formulation:

Figure R4. Calculating the area of each branch in simulation of branching patterns. **A.** A bifurcated branch in a simulated colony. The shaded area represents the branches. Blue lines represent the midlines of the branches and blue dots represent the branch tips. **B.** The two subbranches (Branch 1 and Branch 2) developed from the same branch are highlighted by the orange and the green dashed lines, respectively. **C.** Region a or b belongs only to Branch 1 or Branch 2, and Region c is shared by both branches. Therefore, the area of Branch 1 is $(a + c/2)$ and the area of Branch 2 is $(b + c/2)$.

Figure R4A shows a part of a colony. A branch is defined by the trajectory of a branch tip; if a branch bifurcates, both subbranches inherit the old branch. Figure R4B highlights the two branches in red (Branch 1) and in green (Branch 2), respectively. As shown in Figure R4C, c is a region shared by both branches, while a and b are not. Therefore, the area of Branch 1, $A_1 = a + c/2$, and the area of Branch 2, $A_2 = b + c/2$.

4. In Fig. 1C and in Supplemental Eq.(3), authors used a partial time derivative of C , however since C is the total cell mass which is only a function of time, the derivative should be ordinary.

We thank the reviewer for pointing this out. We corrected this in the revised manuscript.

5. There is an incorrect spelling in the citation (Jacob 2004) In fact, the first author is E. Ben-Jacob.

We thank the reviewer for pointing this out. We corrected this in the revised manuscript.

Reviewer #3:

Theoretical approaches to understanding biological pattern formation typically attempt to explain the bottom-up, mechanistic processes that lead to patterns. This study takes an opposing, top-down approach, finding the patterns that would theoretically optimise growth given a set of environmental conditions and comparing them to those observed experimentally under those conditions. The specific system used to test this conceptual framework is the *P. aeruginosa* swarming motility system, which generates branching patterns as cells move over an agar surface. Using a simple, nutrient-based model of colony growth, the authors predict the yield of colonies with differing branching behaviours and find the geometrical parameters of the branching network (branch density and width) that optimise this yield for a given set of environmental conditions. Experiments suggest that the patterns formed by real colonies in variable environments follow similar trends to the theoretical optimum.

The theory behind this study is elegant, providing an important insight into the ecological constraints on branching pattern formation using a framework that should easily generalize to other systems. Because branching is common to a wide variety of systems other than the *P. aeruginosa* swarming assay used in this study, the results are likely to be of broad interest to a variety of experimental biologists, as well as biophysicists and mathematical biologists. However, I think that the data from the illustrative example of the swarming assay needs to be more carefully analysed to properly support their conclusions.

We appreciate the reviewer for recognizing the significance of the work and for the insightful comments. We address these comments by conducting additional analysis as presented below.

Major comments:

- It should be made much clearer throughout the text that the experimental system is based on the swarming assay. Without this context, statements such as 'Typically, *Pseudomonas* colonies form branches that extends in two dimensions (2D) when initiated from a point inoculation' (line 93) are not true - the branching effect observed in swarming colonies is not typical of many surface-grown *Pseudomonas* colonies, but is observed under a fairly narrow range of experimental conditions (Tremblay, J. and Déziel, E., 2008).

We thank the reviewer for pointing this out. We made revisions throughout the text to emphasize that our experimental system is based on the swarming assay. We have also cited the relevant references the reviewer noted.

- It is hard to tell by eye whether the width of the branches of the colonies in Figs. 2B and 3C is following the predicted trends. The authors' case would be considerably strengthened if they could measure this across all their replicates and present this data alongside the representative images. Quantifying the branch density of the colonies would also be useful.

We thank the reviewer for the suggestion and we agree that our work will be greatly improved by quantification of the experimental data. Other reviewers raised the same point. As the reviewer suggested, we quantified the branch width of the images to support this conclusion and added the following figure to the manuscript (Figure EV2).

Figure R1 shows the predicted optimal branch width (panel A) and the average branch width of *Pseudomonas* (panel B), and they follow a similar trend: the branch width increases with increasing

nutrient concentration or decreasing agar density in the medium. For statistical analysis, we pooled samples grew under similar conditions together (panel C) and calculated the average branch width of four condition groups (panel D), which shows a significant increase in branch width in colonies on rich media or media with low agar densities. In both the model and experimental data, the relationship between the branch density and the nutrient concentration or the expansion efficiency varies and depends on other parameters.

- *P. aeruginosa* can regulate swarming motility according to environmental conditions, changing it based on e.g. the main carbon source available (Shrout et al. 2006), the availability of nitrate (Déziel, E. et al. 2003), or the agar concentration (Kamatkar and Shrout, 2011). However, the modelling approach assumes that the expansion rate of the branches (a proxy for cell motility) is fixed, rather than a physiological parameter that can itself be optimized. While I can appreciate the elegance of restricting the optimisation to the geometrical properties of the branches, the possibility that the trade-off between expansion speed and growth rate may also be a target of evolutionary optimization should at least be mentioned in either the introduction or the discussion.

This is a great point. We agree with the reviewer that other than the colony geometry, many physiological parameters, such as the cell motility, the cell growth rate, the balance between motility and growth, etc., are also subjected to evolutionary optimization. As suggested, we briefly discussed about the other targets of optimization in the Discussion. In this work, we propose that evolution not only selects at the cellular level, but also selects for the optimal colony morphology, which is a macroscopic, population-level property resulted from complex interactions of the cellular-level dynamics. In this light, it is possible that the other cellular level properties are being optimized not independently from each other, but in a coordinated manner to achieve the optimal colony patterns. In our optimization models, we fix the other parameters and only investigate how patterns affect colony growth, in order to find out *with a particular set of physiological properties* (i.e., with a particular level of cell motility), what patterns are optimal for colony growth. Therefore, we focus our analysis on the geometrical properties of the colony, instead of the other physiological parameters.

- Can the use of the cell-density dependent terms (i.e. $cmC/(cm + C)$) in Eq. 2 of the supplement be more thoroughly justified? As far as I can tell, this expression is purely based on the early growth rate data in Supplementary Fig. 1A, which shows slower growth between $t = 0-4$ h. However, this is likely to simply be the lag phase following transferral of cells from stationary-phase culture to fresh media (lines 364-365), rather than a cell density-dependent effect. I would expect that preparing the growth curve plate from an exponential-phase sub-culture rather than the overnight culture would remove this early stage lag.

If these terms are not necessary, I recommend that they be removed for simplicity. The expression for fG will then simplify to the Monod equation, the standard model for the relationship between nutrient concentration and growth rate (e.g. Shluter et al. 2015, Reino et al. 2016).

We thank the reviewer for these suggestions. We indeed have not considered the lag effect in the early stage of our growth curve measurements. As the reviewer suggested, we prepared cell cultures for growth curve measurements from an exponential-phase sub-culture (prepared by diluting overnight cell culture in fresh media and reviving for 3 hours). The growth curves of cells initiated from the exponential-phase show no significant difference from those of cells initiated from the stationary phase (Figure R5). Both are approximately linear at high cell density under high nutrient concentrations.

Therefore, we fit the observed growth dynamics with $f_G = \frac{N}{N+K_N} \frac{C_m}{C+C_m} C$, which approximate to $f_G = \frac{N}{N+K_N} C_m$ when $C \gg C_m$. This equation is an empirical approximation of the observed growth dynamics of *Pseudomonas* cells in swarming media, rather than a description of the underlying mechanisms.

Figure R5. Growth curves of cells initiated from different growth phases. Comparison of growth curves of cells initiated from (A) the stationary phase (overnight cell culture) or (B) the exponential phase (overnight cell culture diluted 200 folds with fresh LB medium and grow for another 3 hours so that OD600 reaches 0.3). To measure the growth curves, 1 μ l cell culture of different growth phases (diluted to the same cell density: OD600 = 0.2) was added to 200 μ l fresh liquid swarming media. Cells were then incubated in a plate reader at 37 °C and OD600 measurements were taken at 10-min intervals for 24 hours.

We also tried removing the cell-density dependent term and compared the results of these two versions (Figure R6). Panel A and B show the results obtained with the cell-density dependent term (as shown in Figure 3A) or with the simplified, Monod equation. Except for some minor differences, the general trends of these two sets of data are consistent, and our conclusions are not affected by this simplification.

Since our subsequent parameter fitting and predictions were performed using the model with the cell-density dependent term, and this term is necessary to approximate the observed growth dynamics of *Pseudomonas* cells in swarming media, we kept the model formulation unchanged in the revised manuscript.

Figure R6. The optimal colony patterns predicted by models with empirical or simplified growth functions. Comparison of the predicted optimal patterns by two different versions of model with the same set of parameters: **A.** The growth function contains a cell density-dependent term to approximate the linear growth curves of *Pseudomonas* in swarming media at high cell density; **B.** The growth function is simplified to the Monod equation. Colors indicate the total biomass at the same time point ($t = 24$ h) scaled to the min/max values in each subpanel.

Minor comments:

Figures:

- A scalebar should be added to at least one of the images in each set of colony images.

We thank the reviewer for pointing this out. We added scalebars to our figures.

- Fig. 2A - colourbars need labels (e.g. 'Biomass', as in Fig. 3)

We thank the reviewer for pointing this out. We added color bar labels to Figure 2A.

- Fig. 2B - are all the images taken at the same time following inoculation? I am confused by sub-panel (c), which seems to show a colony that has expanded less far than the colony in (a) despite apparently being grown under conditions that permit faster expansion. Please make the imaging times clear either in the caption or as part of the figure.

We apologize for not providing the information. Yes, the images of different subpanels were taken at the same time (20 hours after inoculation). We included this information in the figure caption. Although cells travel faster on medium with lower agar density, the colony as a whole may expand slower. This discrepancy probably highlights the impact of patterns on colony growth, as our study

showed, colony pattern is tightly associated with growth efficiency. Indeed, we often observed colonies grew on lower agar density were smaller than those on higher agar density after the same growth time (Figure R7).

Figure R7. *Pseudomonas* colonies grow on swarming media with different agar densities. Concentration of casamino acids: 8 g/L. Images were taken at 18 hours after inoculation.

- Supplementary Fig. 4, condition $D=0.15$ $W=1.5$ - This particular simulation seems to have been buggy, as there are large gaps in the colony perimeter that should have been filled in by the branching algorithm.

Thanks for noticing this. We checked this simulation result and found the large gaps to be a real behavior of the simulated system, amplified by the stochasticity implemented in the model. This happens when the branches are relatively sparse and the branch extension is very fast, so occasionally some branches extend a long distance before they fill in the gap between the branches.

Figure R8 shows four patterns generated with the same set of parameters ($D=0.15$, $W=1.5$). Some patterns have gaps while others do not. As the main purpose of Supplementary Fig 4 (Figure EV3 in the revised manuscript) is to demonstrate how patterns with different combinations of branch widths and densities look like, we replaced this particular simulated pattern with one that does not show large gaps to avoid confusion.

Figure R8. Stochasticity in the simulation of patterns with 2D branch extension. Four examples of simulated patterns with $D=0.15$, $W=1.5$ and the same parameters in Figure EV3. Due to the stochasticity introduced to the branch initiation and branch growth directions, the patterns differ between each simulation.

- Supplementary Fig. 5 - needs colourbar

Thanks for pointing this out. We added color bars to Supplementary Figure 5.

Text:

- I can't find experimental details of how Supplementary Figure 3 was prepared. How does 'solid LB media' differ from the standard swarming media with equal agar concentration?

We added experimental details of how Supplementary Figure 3 (Appendix Figure S2 in the revised manuscript) was prepared to its legends and the Methods (“Measuring the relative mobility of cells”):

“*Pseudomonas* colonies were grown on solid LB media, so they expand radially without developing branches. After 16 hours of growth in 37°C, the diameters of the colonies were measured under a microscope every 2-3 hours. The growth speed (the increasing rate of the colony diameter), v , of each colony is obtained by fitting the diameters to linear functions of time. Assuming the growth and expansion of a radially expanding colony can be described by Fisher’s equation, the cell diffusivity or the relative mobility of cells is proportional to v^2 (Murray, 2007).”

Solid LB media are standard lysogeny broth with agar, and the composition is different from that of the swarming media. To determine how agar density affects the diffusion of bacterial cells, we took advantage of the fact that a radially expanding colony advances as a traveling wave with a speed proportional to \sqrt{D} (D is the diffusivity of cells) (Murray, 2007). By measuring the expanding speed of colonies, we can estimate the relative cell diffusivity. However, since this rule is valid only for radial symmetric colonies, we have to grow colonies on solid LB media, where they do not develop branches, instead of using the swarming media. We do appreciate the fact that the mobility of cells is affected by the medium composition, so cell diffusivity measured on LB media is not the same as on the swarming media. As such, this experiment was intended to measure a *relative* cell diffusivity (comparing between different agar densities) and to show the inverse correlation between cell mobility and the agar densities, we considered the results obtained on LB media a good approximation.

- Lines 143 - 146: Figure callouts of Fig. 2B need to specify the sub-panel referred to.

Thanks for pointing this out. We specified the sub-panel being referred to in the revised manuscript.

- In general, more care should be taken to proof-read the manuscript - e.g. edit line 39-40 to '...which are dwarfed in complexity and sophistication by patterns found in nature.', line 239 to '...more sophisticated descriptions of cell motility have been used..

We thank the reviewer for pointing this out. We made the corrections and checked the grammar of our text throughout.

References

- Ben-Jacob E, Schochet O, Tenenbaum A, Cohen I, Czirok A, Vicsek T (1994) Generic modelling of cooperative growth patterns in bacterial colonies. *Nature* 368: 46-9
- Farrell FD, Hallatschek O, Marenduzzo D, Waclaw B (2013) Mechanically driven growth of quasi-two-dimensional microbial colonies. *Phys Rev Lett* 111: 168101
- Giverso C, Verani M, Ciarletta P (2015) Branching instability in expanding bacterial colonies. *J R Soc Interface* 12: 20141290
- Kawasaki K, Mochizuki A, Matsushita M, Umeda T, Shigesada N (1997) Modeling spatio-temporal patterns generated by *Bacillus subtilis*. *J Theor Biol* 188: 177-85
- Kozlovsky Y, Cohen I, Golding I, Ben-Jacob E (1999) Lubricating bacteria model for branching growth of bacterial colonies. *Phys Rev E Stat Phys Plasmas Fluids Relat Interdiscip Topics* 59: 7025-35
- Matsushita M, Wakita J, Itoh H, Ràfols I, Matsuyama T, Sakaguchi H, Mimura M (1998) Interface growth and pattern formation in bacterial colonies. *Physica A: Statistical Mechanics and its Applications* 249: 517-524
- Mimura M, Sakaguchi H, Matsushita M (2000) Reaction–diffusion modelling of bacterial colony patterns. *Physica A: Statistical Mechanics and its Applications* 282: 283-303
- Murray JD (2007) *Mathematical biology: I. An introduction*. Springer Science & Business Media,
- Trinschek S, John K, Thiele U (2018) Modelling of surfactant-driven front instabilities in spreading bacterial colonies. *Soft Matter* 14: 4464-4476
- van Ditmarsch D, Boyle KE, Sakhtah H, Oyler JE, Nadell CD, Deziel E, Dietrich LE, Xavier JB (2013) Convergent evolution of hyperswarming leads to impaired biofilm formation in pathogenic bacteria. *Cell Rep* 4: 697-708

Thank you again for sending us your revised manuscript. We have now heard back from the three referees who were asked to evaluate the revised study. As you will see below, they are satisfied with the performed revisions and are supportive of publication. Reviewer #3 only lists a couple of suggestions for minor revisions in one of the figures, which we would ask you to address, together with a few remaining editorial issues listed below.

REFEREE REPORTS

Reviewer #1:

The authors have done an outstanding job of addressing all my comments. I also think that they sufficiently addressed the other reviewers' comments. I think the additional controls solidify the already strong conclusions and the edited text make the work more understandable. Especially given COVID, the authors really did an outstanding job in revising their work.

This is really a conceptually novel work done rigorously, both experimentally and theoretically.

Congratulations to the authors on this beautiful work. I support publication without further revisions.

Hyun Youk.

Reviewer #2:

Authors did a good job clarifying the issues I and the other referees raised. I think the paper is acceptable for publication in the present form

Reviewer #3:

Many thanks to the authors for their thorough and considered replies to my comments. I am pleased to see that their new analyses support their hypothesis, and think their central thesis has been considerably strengthened as a result.

In their response, the reviewers state 'In our optimization models, we fix the other parameters and only investigate how patterns affect colony growth, in order to find out with a particular set of physiological properties (i.e., with a particular level of cell motility), what patterns are optimal for colony growth.' I would dispute this apparent view that a given suite of simulations producing different colony geometries is based on a single set of physiological parameters. Even if we are agnostic about the precise mechanisms causing the observed differences in the branching patterns, it seems they must ultimately be physiological in basis for them to be regulated by the cells. However, this is a somewhat tangential point to the manuscript and I don't think any edits need to be made because of it - indeed, to the extent this point comes up in the new version of the manuscript, the authors appear to agree with my view that colony geometry is ultimately physiologically derived.

I have some very minor formatting suggestions for the new figure, but am satisfied that my points have been fully addressed.

Minor comments:

- Fig EV2: Can panels A and B of Fig R1 be moved further apart? It currently looks like the colourbar of A is labelled as 'Agar density' currently.
- On a related note, can the colourbars of A and B be labelled as 'optimal branch width' and 'average branch width', respectively?

The authors have made all requested editorial changes.

ACCEPTED

15th Mar 2021

Thank you for sending us your revised manuscript and for performing the last requested text edits. I am pleased to inform you that your paper has been accepted for publication.

YOU MUST COMPLETE ALL CELLS WITH A PINK BACKGROUND ↓
PLEASE NOTE THAT THIS CHECKLIST WILL BE PUBLISHED ALONGSIDE YOUR PAPER

Corresponding Author Name: Lingchong You
Journal Submitted to: Molecular Systems Biology
Manuscript Number: MSB-2020-10089